# Walking Out of the Weisfeiler Leman Hierarchy: Graph Learning Beyond Message Passing

**Jan Tönshoff**                                                            *toenshoff@informatik.rwth-aachen.de*
*RWTH Aachen University*

**Martin Ritzert**                                                          *ritzert@informatik.uni-goettingen.de*
*Georg-August-Universität Göttingen*

**Hinrikus Wolf**                                                           *hinrikus@cs.rwth-aachen.de*
*RWTH Aachen University*

**Martin Grohe**                                                            *grohe@informatik.rwth-aachen.de*
*RWTH Aachen University*

**Reviewed on OpenReview:** *https://openreview.net/forum?id=vgXnEyeWVY*

## Abstract

We propose CRAWL, a novel neural network architecture for graph learning. Like graph neural networks, CRAWL layers update node features on a graph and thus can freely be combined or interleaved with GNN layers. Yet CRAWL operates fundamentally different from message passing graph neural networks. CRAWL layers extract and aggregate information on subgraphs appearing along random walks through a graph using 1D Convolutions. Thereby, it detects long range interactions and computes non-local features. As the theoretical basis for our approach, we prove a theorem stating that the expressiveness of CRAWL is incomparable with that of the Weisfeiler Leman algorithm and hence with graph neural networks. That is, there are functions expressible by CRAWL, but not by GNNs and vice versa. This result extends to higher levels of the Weisfeiler Leman hierarchy and thus to higher-order GNNs. Empirically, we show that CRAWL matches state-of-the-art GNN architectures across a multitude of benchmark datasets for classification and regression on graphs.

## 1 Introduction

Over the last five-years, graph learning has become dominated by graph neural networks (GNNs), unquestionably for good reasons. Yet GNNs do have their well-known limitations, and an important research direction in recent years has been to extend the expressiveness of GNNs without sacrificing their efficiency. Instead, we propose a novel neural network architecture for graphs called CRAWL (**C**NNs for **Ra**ndom **W**alks). While still being fully compatible with GNNs in the sense that, just like GNNs, CRAWL iteratively computes node features and thus can potentially be integrated into a GNN architecture, CRAWL is based on a fundamentally different idea, and the features it extracts are provably different from those of GNNs. CRAWL samples a set of random walks and extracts features that fully describe the subgraphs visible within a sliding window over these walks. The walks with the subgraph features are then processed with standard 1D convolutions. The outcome is pooled into the nodes such that each layer updates a latent embedding for each node, similar to GNNs and graph transformers.

The CRAWL architecture was originally motivated from the empirical observation that in many application scenarios random walk based methods perform surprisingly well in comparison with graph neural networks. A notable example is node2vec (Grover & Leskovec, 2016) in combination with various classifiers. A second observation is that standard GNNs are not very good at detecting small subgraphs, for example, cycles of length 6 (Morris et al., 2019; Xu et al., 2019). The distribution of such subgraphs in a graph carries relevant information about the structure of a graph, as witnessed by the extensive research on motif detection and counting (e.g. Alon, 2007). CRAWL detects both the global connectivity structure in a graph by sampling longer random walks as well as the full local structure and hence all subgraphs within its window size. We carry out a comprehensive theoretical analysis of the expressiveness of CRAWL. Our main theorem shows that CRAWL is incomparable with GNNs in terms of expressiveness: (1) there are graphs that can be distinguished by CRAWL, but not by GNNs, and (2) there are graphs that can be distinguished by GNNs, but not by CRAWL. The first result is based on the well-known characterization of the expressiveness of GNNs in terms of the Weisfeiler-Leman algorithm (Morris et al., 2019; Xu et al., 2019). Notably, the result can be extended to the full Weisfeiler Leman hierarchy and hence to higher-order GNNs: for every $k$ there are graphs that can be distinguished by CRAWL, but not by $k$-dimensional Weisfeiler Leman. In particular, this means that the result also applies to the variety of GNN-extensions whose expressiveness is situated within the Weisfeiler Leman hierarchy (e.g. Barceló et al., 2021; Cotta et al., 2021; Maron et al., 2019a).

Yet it is not only these theoretical results making CRAWL attractive. *We believe that the key to the strength of CRAWL is a favorable combination of engineering and expressiveness aspects.* The parallelism of modern hardware allows us to efficiently sample large numbers of random walks and their feature matrices directly on the GPU. Once the random walks are available, we can rely on existing highly optimized code for 1D CNNs, again allowing us to exploit the strength of modern hardware. Compared to other methods based on counting or sampling small subgraphs, for example graphlet kernels (Shervashidze et al., 2009b), CRAWL's method of sampling small subgraphs in a sliding window on random walks has the advantage that even in sparse graphs it usually yields meaningful subgraph patterns. Its way of encoding the local graph structure and processing this encoding gives CRAWL an edge over subgraph GNNs (Frasca et al., 2022) and random-walk-based neural networks such as RAW-GNN (Jin et al., 2022).

We carry out a comprehensive experimental analysis, demonstrating that empirically CRAWL is on par with advanced message passing GNN architectures and graph transformers on major graph learning benchmark datasets and excels when it comes to long-range interactions.

## 1.1 Related Work

Message passing GNNs (MPGNNs) (Gilmer et al., 2020), as the currently dominant graph learning architecture, constitute the main baselines in our experiments. Many variants of this architecture exist (see Wu et al., 2020), such as GCN (Kipf & Welling, 2017), GIN (Xu et al., 2019), GAT (Veličković et al., 2018), GraphSage (Hamilton et al., 2017), and PNA (Corso et al., 2020). Multiple extensions to the standard message passing framework have been proposed that strengthen the theoretical expressiveness which otherwise is bounded by the 1-dimensional Weisfeiler-Leman algorithm. With 3WLGNN, Maron et al. (2019a) suggested a higher-order GNN, which is equivalent to the 3-dimensional Weisfeiler-Leman kernel and thus more expressive than standard MPGNNs. In HIMP (Fey et al., 2020), the backbone of a molecule graph is extracted and then two GNNs are run in parallel on the backbone and the full molecule graph. This allows HIMP to detect structural features that are otherwise neglected. Explicit counts of fixed substructures such as cycles or small cliques have been added to the node and edge features by Bouritsas et al. (2020) (GSN). Similarly, Sankar et al. (2017), Lee et al. (2019), and Peng et al. (2020) added the frequencies of motifs, i.e., common connected induced subgraphs, to improve the predictions of GNNs. Sankar et al. (2020) introduce motif-based regularization, a framework that improves multiple MPGNNs. In (Jiang et al., 2022) the authors introduce sparse motifs which help to reduce over-smoothing in their architecture called Sparse-Motif Ensemble Graph Convolutional Network. Another approach with strong empirical performance is GINE+ (Brossard et al., 2020). It is based on GIN and aggregates information from higher-order neighborhoods, allowing it to detect small substructures such as cycles. Beaini et al. (2021) proposed DGN, which incorporates directional awareness into message passing. Bodnar et al. (2021) propose CW Networks, which incorporate regular Cell Complexes into GNNs to construct architectures that are not less powerful than 3-WL.

A different way to learn on graph data is to use similarity measures on graphs with graph kernels (Kriege et al., 2020). Graph kernels often count induced subgraphs such as graphlets, label sequences, or subtrees, which relates them conceptually to our approach. The graphlet kernel (Shervashidze et al., 2009a) counts the occurrences of all 5-node (or more general $k$-node) subgraphs. The Weisfeiler-Leman kernel (Shervashidze et al., 2011) is based on iterated degree sequences and effectively counts occurrences of local subtrees. A slightly more expressive graph kernel based on a local version of a higher-dimensional Weisfeiler-Leman algorithm was introduced in (Morris et al., 2020b). The Weisfeiler-Leman algorithm is the traditional yardstick for the expressiveness of GNN architectures.

Another line of research studies Graph Transformers Dwivedi & Bresson (2020) based on Transformer networks (Vaswani et al., 2017). By combining the spectral decomposition of the graph Laplacian with global transformer-based attention, spectral attention networks (SAN) by Kreuzer et al. (2021) focus on global patterns in graphs. Chen et al. (2022) propose Structure-Aware Transformers (SAT), which embed all $k$-hop subgraphs in a graph with a GNN and then process the resulting set of subgraph embeddings with a Transformer. In contrast to SAN, SAT thus focuses more on local features, similar to MPGNNs.

A few previous approaches utilize conventional CNNs in the context of end-to-end graph learning. Patchy-SAN (Niepert et al., 2016) normalizes graphs in such a way that they can be interpreted by CNN layers. Zhang et al. (2018) proposed a pooling strategy based on sorting intermediate node embeddings and presented DGCNN which applies a 1D CNN to the sorted embeddings of a graph. Yuan & Ji (2021) used a 1D CNN layer and attention for neighborhood aggregation to compute node embeddings.

Several architectures based on random walks have been proposed. Nikolentzos & Vazirgiannis (2020) propose a differentiable version of the random walk kernel and integrate it into a GNN architecture where they learn a number of small graph against which the differentiable random walk kernel compares the nodes of the graph. In (Geerts, 2020), the $\ell$-walk MPGNN adds random walks directly as features to the nodes and connects the architecture theoretically to the 2-dimensional Weisfeiler-Leman algorithm. RAW-GNN (Jin et al., 2022) and pathGCN (Eliasof et al., 2022) aggregate node features along random walks, learning to aggregate information based on the distance to the start of the walk. This aggregation allows RAW-GNN to tackle node predictions in a heterophilic domain. Similar distance-based aggregations are also possible in a multi-hop MPGNN by Ma et al. (2020). Note that these approaches do not encode the structure traversed by the walk at all, instead the methods focus solely on proximity. In contrast, CRAWL explicitly extracts and uses local structure.

Ivanov & Burnaev (2018) proposed a structure-aware approach by encoding node identity along anonymous random walks to learn graph representations for SVMs. Wang et al. (2021) extend the concept of anonymous random walks to temporal graphs. They also use encodings of node identity as features which are processed by an RNN to solve temporal link prediction tasks. In a related approach, Yin et al. (2022) learn subgraph representations based on random walks sampled around a set of query vertices. Their features are based on histograms of walk distances and are also processed by RNNs. They apply their approach to link prediction tasks and demonstrate performance superior to standard MPGNNs. The previous two methods share conceptual parallels with CRAWL, but there are several core differences: The features of CRAWL encode both the identity and adjacency between vertices (instead of just identity) to provide an explicit and complete encoding of subgraph structure. Furthermore, our approach does not just compute an embedding of a subgraph around query vertices. Instead, we use CRAWL to embed all vertices at once with a more parallelizable CNN in a manner that is compatible with message passing, Virtual Nodes, and other graph-learning layers based on iteratively refining vertex embeddings. These differences enable CRAWL to tackle tasks where local structural features are of high importance and a bottleneck of standard GNN architectures such as graph classification in the molecular domain.

Through its encoding of the complete local structure around a random walk, CRAWL processes induced subgraphs, and in that regard is similar to Subgraph GNNs (Frasca et al., 2022) and Structure-Aware Transformers (SAT) (Chen et al., 2022). Prior work in this framework includes Nested Graph Neural Networks (Zhang & Li, 2021), Equivariant Subgraph Aggregation Networks (Bevilacqua et al., 2022), and "GNN As Kernel" (Zhao et al., 2022). The main difference to our model is the way local structure is made available to the model. Both Subgraph GNNs and SAT use a standard MPGNN to process subgraphs, therefore keeping some limitations although they are more expressive than 1-WL. In contrast, CRAWL

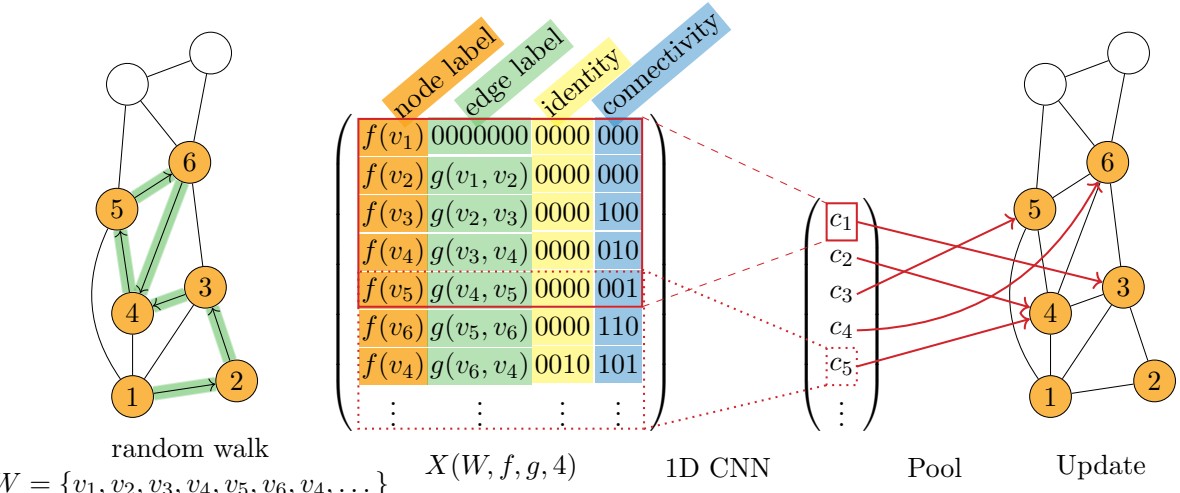

Figure 1: Example of the information flow in a CRAWL layer for a graph with 8 nodes. We sample a walk $W$ and compute the feature matrix $X$ based on node embeddings $f$, edge embeddings $g$, and a window size of $s=4$. To this matrix we apply a 1D CNN with receptive field $r=5$ and pool the output into the nodes to update their embeddings.

defines binary encodings of the (fixed-size) subgraph structure which are then embedded by an MLP which in theory can encode every function. On the other hand, our subgraph representation is not deterministic since it depends on the order in which the random walk traverses the subgraph, while always providing complete structural information. Also, the chosen subgraph is random, depending on the concrete random walk. Due to those differences the expressiveness of MPGNNs and CRAWL is incomparable as formally shown in Section 3.

## 2    Method

CRAWL processes random walks with convolutional neural networks to extract information about local and global graph structure. For a given input graph, we initially sample a large set of (relatively long) random walks. For each walk, we construct a sequential feature representation which encodes the structure of all contiguous walk segments up to a predetermined size (walklets). These features are well suited for processing with a 1D CNN, which maps each walklet to an embedding. These walklet embeddings are then used to update latent node states. By repeating this process, we allow long-range information about the graph structure to flow along each random walk.

### 2.1    Random Walk Features

A walk $W$ of length $\ell \in \mathbb{N}$ in a graph $G = (V, E)$ is a sequence of nodes $W = (v_0, \ldots, v_\ell) \in V^{\ell+1}$ with $v_{i-1}v_i \in E$ for all $i \in [\ell]$. A random walk in a graph is obtained by starting at some initial node $v_0 \in V$ and iteratively sampling the next node $v_{i+1}$ randomly from the neighbors $N_G(v_i)$ of the current node $v_i$. In our main experiments, we choose the next node uniformly among $N_G(v_i) \backslash \{v_{i-1}\}$, i.e. following a non-backtracking uniform random walk strategy. In general, the applied random walk strategy is a hyperparameter of crawl and we perform an ablation study on their effectiveness in Section 5. There, we compare uniform random walks with and without backtracking. Another common random walk strategy are pq-Walks as used in node2vec (Grover & Leskovec, 2016).

We call contiguous segments $W[i:j] := (w_i, \ldots, w_j)$ of a walk $W = (w_0, \ldots, w_\ell)$ *walklets*. The *center* of a walklet $(w_i, \ldots, w_j)$ of even length $j - i$ is the node $w_{(i+j)/2}$. For each walklet $w = (w_i, \ldots, w_j)$, we denote by $G[w]$ the subgraph induced by $G$ on the set $\{w_i, \ldots, w_j\}$. Note that $G[w]$ is connected as it contains all

edges $w_k w_{k+1}$ for $i \le k < j$, and it may contain additional edges. Also note that the $w_k$ are not necessarily distinct, especially if backtracking uniform random walks are used.

For a walk $W$ and a local window size $s$, we construct a matrix representation that encodes the structure of all walklets of size $s+1$ in $W$ as well as the node and edge features occurring along the walk. Given a walk $W \in V^\ell$ of length $\ell - 1$ in a graph $G$, a $d$-dimensional node embedding $f : V \to \mathbb{R}^d$, a $d'$-dimensional edge embedding $g : E \to \mathbb{R}^{d'}$, and a local window size $s > 0$ we define the *walk feature matrix* $X(W, f, g, s) \in \mathbb{R}^{\ell \times d_X}$ with feature dimension $d_X = d + d' + s + (s-1)$ as

$$X(W, f, g, s) = (f_W \ g_W \ I_W^s \ A_W^s).$$

For ease of notation, the first dimensions of each of the matrices $f_W, g_W, I_W^s, A_W^s$ is indexed from 0 to $\ell - 1$. Here, the *node feature sequence* $f_W \in \mathbb{R}^{\ell \times d}$ and the *edge feature sequence* $g_W \in \mathbb{R}^{\ell \times d'}$ are defined as the concatenation of node and edge features, respectively. Formally,

$$(f_W)_{i,\_} = f(v_i) \qquad \text{and} \qquad (g_W)_{i,\_} = \begin{cases} \mathbf{0}, & \text{if } i = 0 \\ g(v_{i-1}v_i), & \text{else.} \end{cases}$$

We define the *local identity relation* $I_W^s \in \{0,1\}^{\ell \times s}$ and the *local adjacency relation* $A_W^s \in \{0,1\}^{\ell \times (s-1)}$ as

$$(I_W^s)_{i,j} = \begin{cases} 1, & \text{if } i-j \ge 0 \wedge v_i = v_{i-j} \\ 0, & \text{else} \end{cases} \qquad \text{and}$$

$$(A_W^s)_{i,j} = \begin{cases} 1, & \text{if } i-j \ge 1 \wedge v_i v_{i-j-1} \in E \\ 0, & \text{else.} \end{cases}$$

The remaining matrices $I_W^s$ and $A_W^s$ are binary matrices that contain one row for every node $v_i$ in the walk $W$. The bitstring for $v_i$ in $I_W^s$ encodes which of the $s$ predecessors of $v_i$ in $W$ are identical to $v_i$, that is, where the random walk looped or backtracked. Micali & Zhu (2016) showed that from the distribution of random walks of bounded length with such identity features, one can reconstruct any graph locally (up to some radius $r$ depending on the length of the walks). Similarly, $A_W^s$ stores which of the predecessors are adjacent to $v_i$, indicating edges in the induced subgraph. The immediate predecessor $v_{i-1}$ is always adjacent to $v_i$ and is thus omitted in $A_W^s$. Note that we do not leverage edge labels of edges not on the walk, only the existence of such edges within the local window is encoded in $A_W^s$.

For any walklet $w = W[i : i+s]$, the restriction of the walk feature matrix to rows $i, \ldots, i+s$ contains a full description of the induced subgraph $G[w]$. Hence, when we apply a CNN with receptive field of size at most $s+1$ to the walk feature matrix, the CNN filter has full access to the subgraph induced by the walklet within its scope. Figure 1 depicts an example of a walk feature matrix and its use in a CRAWL layer.

CRAWL initially samples $m$ random walks $\Omega = \{W_1, \ldots, W_m\}$ of length $\ell$ from the input graph. By stacking the individual feature matrices for each walk, we get the *walk feature tensor* $X(\Omega, f, g, s) \in \mathbb{R}^{m \times \ell \times d_X}$ defined as

$$X(\Omega, f, g, s)_i = X(W_i, f, g, s).$$

This stack of walk feature matrices can then be processed in parallel by a CNN. The values for $m$ and $\ell$ are not fixed hyperparameters of the model but instead can be chosen at runtime. By default, we start one walk at every node, i.e., $m = |V|$. We noted that reducing the number of walks during training can help against overfitting and of course is a way to reduce the memory footprint which is important for large graphs. If we choose to use fewer random walks, we sample $m$ starting nodes uniformly at random. We always make sure that $m\ell \gg |V|$, ensuring that with high probability each node appears on multiple paths. We typically choose $\ell \ge 50$ during training. During inference, we choose a larger $\ell$ of up to 150 which improves the predictions. The window size $s$ has to be at least 6 for molecular datasets to capture organic rings. By default we thus use $s = 8$ and expand the window size to $s = 16$ when handling long-range dependencies.

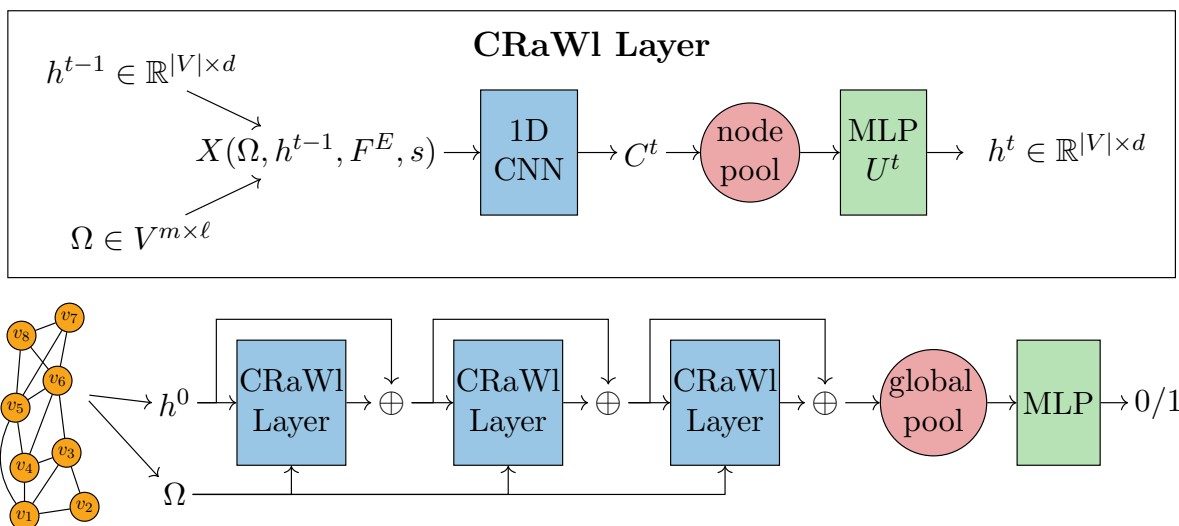

Figure 2: Top: Update procedure of latent node embeddings $h^t$ in a CRAWL layer. $\Omega$ is a set of random walks. Bottom: Architecture of a 3-layer CRAWL network as used in the experiments.

## 2.2 Architecture

A CRAWL network iteratively updates latent embeddings for each node. Let $G = (V, E)$ be a graph with initial node and edge feature maps $F^V : V \to \mathbb{R}^{d_V}$ and $F^E : E \to \mathbb{R}^{d_E}$. The function $h^{(t)} : V \to \mathbb{R}^{d^{(t)}}$ stores the output of the $t$-th layer of CRAWL and the initial node features are stored in $h^{(0)} = F^V$. In principle, the output dimension $d^{(t)}$ is an independent hyperparameter for each layer. In practice, we use the same size $d$ for the output node embeddings of all layers for simplicity.

The $t$-th layer of a CRAWL network constructs the walk feature tensor $X^{(t)} = X(\Omega, h^{(t-1)}, F^E, s)$ using $h^{(t-1)}$ as its input node embedding and the graph's edge features $F^E$. This walk feature tensor is then processed by a convolutional network $\text{CNN}^{(t)}$ based on 1D CNNs. The first dimension of $X^{(t)}$ of size $m$ is viewed as the batch dimension. The convolutional filters move along the second dimension (and therefore along each walk) while the third dimension contains the feature channels. Each $\text{CNN}^{(t)}$ consists of three convolutional layers combined with ReLU activations and batch normalization. A detailed description is provided in Appendix B. The stack of operations has a receptive field of $s+1$ and effectively applies an MLP to each subsection of this length in the walk feature matrices. In each $\text{CNN}^{(t)}$, we use *Depthwise Separable Convolutions* (Chollet, 2017) for efficiency.

The output of $\text{CNN}^{(t)}$ is a tensor $C^{(t)} \in \mathbb{R}^{m \times (\ell - s) \times d}$. Note that the second dimension is $\ell - s$ instead of $\ell$ as no padding is used in the convolutions. Through its receptive field, the CNN operates on walklets of size $s+1$ and produces embeddings for those. We pool those embeddings into the nodes of the graph by averaging for each node $v \in V$ all embeddings of walklets centered at $v$. Let $\Omega = \{W_1, \ldots, W_m\}$ be a set of walks with $W_i = (v_{i,1}, \ldots, v_{i,\ell})$. Then $C^{(t)}_{i,j-s/2} \in \mathbb{R}^d$ is the embedding computed by $\text{CNN}^{(t)}$ for the walklet $w = W_i[j - \frac{s}{2} : j + \frac{s}{2}]$ centered at $v_{i,j}$ and the pooling operation is given by

$$p^{(t)}(v) = \underset{(i,j) \in \text{center}(\Omega, s, v)}{\text{mean}} C^{(t)}_{i,j-s/2} .$$

Here, $\text{center}(\Omega, s, v)$ is the set of walklets of length $s+1$ in $\Omega$ in which $v$ occurs as center. We restrict $s$ to be even such that the center is always well-defined. The output of the pooling step is a vector $p^{(t)}(v) \in \mathbb{R}^d$ for each $v$. This vector is then processed by a trainable MLP $U^{(t)}$ with a single hidden layer of dimension $2d$ to compute the next latent vertex embedding

$$h^{(t)}(v) = U^{(t)}\big(p^{(t)}(v)\big).$$

Figure 2 (top) gives an overview over the elements of a CRAWL layer. Our architecture stacks multiple CRAWL layers into one end-to-end trainable neural network, as illustrated in Figure 2 (bottom). After the final layer we apply batch normalization before performing a global readout step with either sum- or mean-pooling to obtain a graph-level latent embedding. Finally, a simple feedforward neural network is used to produce a graph-level output which can then be used in classification and regression tasks.

Since CRAWL layers are based on iteratively updating latent node embeddings, they are fully compatible with conventional message passing layers and related techniques such as virtual nodes (Gilmer et al., 2017; Li et al., 2017; Ishiguro et al., 2019). In our experiments, we use virtual nodes whenever this increases validation performance. A detailed explanation of our virtual node layer is provided in Appendix A.5. Combining CRAWL with message passing layers is left as future work.

We implemented CRAWL in PyTorch (Paszke et al., 2019; Fey & Lenssen, 2019)[1]. Note that for a given graph we sample all $m$ walks in parallel with a GPU-accelerated random walk generator. This process is fast enough to be done on-the-fly for each new batch instead of using precomputed random walks. Roughly 10% of the total runtime during training is spent on sampling walks.

**Asymptotic Runtime**

We now compare the asymptotic runtime of CRAWL to that of expressive MPGNNs and graph transformers. Neglecting the hidden dimension $d$, the runtime of applying the CNN in each CRAWL layer is $\mathcal{O}(m \cdot \ell \cdot s^2)$. The window size $s$ is squared since it determines both the size of the walk features but also the receptive field of the CNN. The walk feature tensor uses memory in $\mathcal{O}(m \cdot \ell \cdot (s + d))$. Our default choice for training is $m = |V|$ and we usually choose $\ell$ such that $|V| \cdot \ell > |E|$. Therefore, the required time exceeds that of a standard MPGNN layer running in $\mathcal{O}(|V| + |E|)$. Let us emphasize that this is a common drawback of architectures which are not bounded by 1-WL. For example, precomputing structural features with GSN (Bouritsas et al., 2020) has a worst-case runtime in $\mathcal{O}(|V|^k)$, where $k$ is the size of the considered subgraphs. Similarly, higher-order $k$-GNNs (Morris et al., 2020b) have a runtime in $\mathcal{O}(|V|^k)$ while methods based on Graph Transformers (Dwivedi & Bresson, 2020) usually have a time and space requirement in $\mathcal{O}(|V|^2)$. While random node initialization (Abboud et al., 2021; Sato et al., 2020) also strengthens the expressive power of MPGNNs beyond 1-WL, its empirical advantage is limited. With default values for $m, \ell$, and constant $s \approx 10$, CRAWL scales with $\mathcal{O}(|E|)$, such that its complexity is comparable with standard MPGNN layers and smaller than both higher-order $k$-GNNs and graph transformers. Note though that the quadratic dependence on $s$ will make CRAWL slower than standard MPGNNs in practice. We provide runtime measurements in Table 8 in the appendix showing that CRAWL is mostly slower than simple message passing architectures and faster than the graph transformer SAN.

## 3 Expressiveness

In this section, we report on theoretical results comparing the expressiveness of CRAWL with that of message passing graph neural networks. We prove that the expressiveness of the two architectures is incomparable, confirming our intuition that CRAWL detects fundamentally different features of graphs than GNNs. This gives a strong theoretical argument for the use of CRAWL as an alternative graph learning architecture, ideally in combination with GNNs to get the best of both worlds.

Intuitively, the additional strength of CRAWL has two different roots:

(i) CRAWL detects small subgraphs (of size determined by the window size hyperparameter $s$), whereas GNNs (at least in their basic form) cannot even detect triangles;

(ii) CRAWL detects long range interactions (determined by the window size and path length hyperparameters), whereas GNNs are local with a locality radius determined by the number of layers (which is typically smaller).

---

[1] https://github.com/toenshoff/CRaWl

While (ii) does play a role in practice, in our theoretical analysis we focus on (i). This has the additional benefit that our results are independent of the number of GNN layers, making them applicable even to recurrent GNNs where the number of iterations can be chosen at runtime. Let us remark that (i) makes CRAWL similar to network analysis techniques based on motif detection (Alon, 2007) and graph kernels based on counting subgraphs, such as the graphlet kernel (Shervashidze et al., 2009a) (even though the implementation of these ideas is very different in CRAWL).

Conversely, the intuitive advantage GNNs have over CRAWL is that locally, GNNs perform a breadth-first search, whereas CRAWL searches along independent walks. For example, GNNs immediately detect the degree of every node, and then on subsequent layers, the degree distributions of the neighbors, and their neighbors, et cetera. While in principle, CRAWL can also detect degrees by walks that are returning to the same node multiple times, it becomes increasingly unlikely and in fact impossible if the degree is larger than the walk length. Interestingly, large degrees are not the only features that GNNs detect, but CRAWL does not. We separate GNNs from CRAWL even on graphs of maximum degree 3.

It is known that the expressiveness of GNNs corresponds exactly to that of the 1-dimensional Weisfeiler-Leman algorithm (1-WL) (Morris et al., 2019; Xu et al., 2019), in the sense that two graphs are distinguished by 1-WL if and only if they can be distinguished by a GNN. It is also known that higher-dimensional versions of WL characterize the expressiveness of higher-order GNNs (Morris et al., 2019). Our main theorem states that the expressiveness of CRAWL and $k$-WL is incomparable.

**Theorem 1.**

*(1) For every $k \geq 1$ there are graphs of maximum degree 3 that are distinguishable by CRAWL with window size and walk length $O(k^2)$, but not by $k$-WL (and hence not by $k$-dimensional GNNs).*

*(2) There are graphs of maximum degree 3 that are distinguishable by 1-WL (and hence by GNNs), but not by CRAWL with window size and path length $O(n)$, where $n$ is the number of vertices of the graph.*

We formally prove this theorem in Sections 3.1 and 3.2. To make the theorem precise, we first need to define what it actually means that CRAWL, as a randomized algorithm, *distinguishes* two graphs. To this end, we give a formal definition based on the total variation distance of the distribution of walk feature matrices that we observe in the graphs. Our proof of assertion (1) assumes that the CNN filters used to process the features within a window on a walk can be arbitrary multilayer perceptrons and hence share their universality. Once this is settled, we can prove (1) by using the seminal Cai-Fürer-Immerman construction (Cai et al., 1992) providing families of graphs indistinguishable by the WL algorithm. The proof of (2) is based on an analysis of random walks on a long path based on the gambler's ruin problem. Before we give the proof, let us briefly comment on the necessary window size and simple extensions of the theorem.

In practice, graph neural networks of order at most $k = 3$ are feasible. Thus, the window size and path length $O(k^2)$ required by CRAWL models to exceed the expressiveness of GNNs of that order (according to Assertion (1) of the theorem) is very moderate. It can also be shown that CRAWL with a window size polynomial in the size of the graphlets is strictly more expressive than graphlet kernels. This can be proved similarly to Assertion (1) of Theorem 1.

Let us remark that the expressiveness of GNNs can be considerably strengthened by adding a random node initialization (Abboud et al., 2021; Sato et al., 2020). The same can be done for CRAWL, but so far the need for such a strengthening (at the cost of a higher variance) did not arise. Besides expressiveness, a fundamental aspect of the theory of machine learning on graphs is invariance (Maron et al., 2019b). So let us briefly comment on the invariance of CRAWL computations. A CRAWL model represents a random variable mapping graphs into a feature space $\mathbb{R}^d$, and this random variable is invariant, which means that it only depends on the isomorphism type of the input graph. By independently sampling more walks we can reduce the variance of the random variable and converge to the invariant "true" function value. Note that this invariance of CRAWL is the same as for other graph learning architectures, in particular GNNs with random node initialization (see Abboud et al., 2021).

**Notation**   Let us first recall the setting and introduce some additional notation. Throughout the proofs, we assume that the graphs are undirected, simple (that is, without self-loops and parallel edges) and unlabeled. All results can easily be extended to (vertex and edge) labeled graphs.[2] In fact, the (harder) inexpressivity results only become stronger by restricting them to the subclass of unlabeled graphs. We further assume that graphs have no isolated nodes, which enables us to start a random walk from every node. This makes the setup cleaner and avoids tedious case distinctions, but again is no serious restriction.

We denote the node set of a graph $G$ by $V(G)$ and the edge set by $E(G)$. The *order* $|G|$ of $G$ is the number of nodes, that is, $|G| := |V(G)|$. For a set $X \subseteq V(G)$, the *induced subgraph* $G[X]$ is the graph with node set $X$ and edge set $\{vw \in E(G) \mid v, w \in X\}$. A walk of length $\ell$ in $G$ is a sequence $W = (w_0, \ldots, w_\ell) \in V(G)^{\ell+1}$ such that $w_{i-1}w_i \in E(G)$ for $1 \le i \le \ell$. The walk is *non-backtracking* if for $1 < i < \ell$ we have $w_{i+1} \ne w_{i-1}$ unless the degree of vertex $w_i$ is 1.

**Distinguishing Graphs**   Before we prove the theorem, let us precisely specify what it means that CRAWL distinguishes two graphs. Recall that CRAWL has three (walk related) hyperparameters:

- the *window size $s$*;

- the *walk length $\ell$*;

- the *sample size $m$*.

Recall furthermore that with every walk $W = (w_0, \ldots, w_\ell)$ we associate a *walk feature matrix* $X \in \mathbb{R}^{(\ell+1) \times (d+d'+s+(s-1))}$. For $0 \le i \le \ell$, the first $d$ entries of the $i$-th row of $X$ describe the current embedding of the node $w_i$, the next $d'$ entries the embedding of the edge $w_{i-1}w_i$ (0 for $i = 0$), the following $s$ entries are indicators for the equalities between $w_i$ and the nodes $w_{i-j}$ for $j = 1, \ldots, s$ (1 if $w_i = w_{i-j}$, 0 if $i - j < 0$ or $w_i \ne w_{i-j}$), and the remaining $s - 1$ entries are indicators for the adjacencies between $w_i$ and the nodes $w_{i-j}$ for $j = 2, \ldots, s$ (1 if $w_i, w_{i-j}$ are adjacent in $G$, 0 if $i - j < 0$ or $w_i, w_{i-j}$ are non-adjacent; note that $w_i, w_{i-1}$ are always be adjacent because $W$ is a walk in $G$). Since on unlabeled graphs initial node and edge embeddings are the same for all nodes and edges by definition, those embeddings cannot contribute to expressivity. Thus, we can safely ignore these embeddings and focus on the subgraph features encoded in the last $2s - 1$ columns. For simplicity, we regard $X$ as an $(\ell + 1) \times (2s - 1)$ matrix with only these features in the following. We denote the entries of the matrix $X$ by $X_{i,j}$ and the rows by $X_{i,-}$. So $X_{i,-} = (X_{i,1}, \ldots, X_{i,2s-1}) \in \{0,1\}^{2s-1}$. We denote the walk feature matrix of a walk $W$ in a graph $G$ by $X(G, W)$. The following observation formalizing the connection between walk-induced subgraphs and rows in the matrix $X$ is immediate from the definitions.

**Observation 1.** *For all walks $W = (w_1, \ldots, w_\ell), W' = (w'_1, \ldots, w'_\ell)$ in graphs $G, G'$ with feature matrices $X := X(G, W), X' := X(G', W')$, we have:*

(1) *if $X_{i-j,-} = X'_{i-j,-}$ for $j = 0, \ldots, s - 1$ then the mapping $w_{i-j} \mapsto w'_{i-j}$ for $j = 0, \ldots, s$ is an isomorphism from the induced subgraph $G[\{w_{i-j} \mid j = 0, \ldots, s\}]$ to the induced subgraph $G'[\{w'_{i-j} \mid j = 0, \ldots, s\}]$;*

(2) *if the mapping $w_{i-j} \mapsto w'_{i-j}$ for $j = 0, \ldots, 2s - 1$ is an isomorphism from the induced subgraph $G[\{w_{i-j} \mid j = 0, \ldots, 2s - 1\}]$ to the induced subgraph $G'[\{w'_{i-j} \mid j = 0, \ldots, 2s - 1\}]$, then $X_{i-j,-} = X'_{i-j,-}$ for $j = 0, \ldots, s - 1$.*

The reason that we need to include the vertices $w_{i-2s+1}, \ldots, w_{i-s}$ and $w'_{i-2s+1}, \ldots, w'_{i-s}$ into the subgraphs in (2) is that row $X_{i-s+1,-}$ of the feature matrix records edges and equalities between $w_{i-s+1}$ and $w_{i-2s+1}, \ldots, w_{i-s}$.

For every graph $G$ we denote the distribution of random walks on $G$ starting from a node chosen uniformly at random by $\mathcal{W}(G)$ and $\mathcal{W}_{nb}(G)$ for the non-backtracking walks. We let $\mathcal{X}(G)$ and $\mathcal{X}_{nb}(G)$ be the push-forward

---

[2]It is possible to simulate directed edges and parallel edges through edge labels and loops through node labels.

distributions on $\{0,1\}^{(\ell+1)\times(2s-1)}$, that is, for every $X \in \{0,1\}^{(\ell+1)\times(2s-1)}$ we let

$$\Pr_{\mathcal{X}(G)}(X) = \Pr_{\mathcal{W}(G)}\big(\{W \mid X(G,W) = X\}\big)$$

A CRAWL run on $G$ takes $m$ samples from $\mathcal{X}(G)$. So to distinguish two graphs $G, G'$, CRAWL must detect that the distributions $\mathcal{X}(G), \mathcal{X}(G')$ are distinct using $m$ samples.

As a warm-up, let us prove the following simple result.

**Theorem 2.** *Let $G$ be a cycle of length $n$ and $G'$ the disjoint union of two cycles of length $n/2$. Then $G$ and $G'$ cannot be distinguished by CRAWL with window size $s < n/2$ (for any choice of parameters $\ell$ and $m$).*

*Proof.* With a window size smaller than the length of the shortest cycle, the graph CRAWL sees in its window is always a path. Thus, for every walk $W$ in either $G$ or $G'$ the feature matrix $X(W)$ only depends on the backtracking pattern of $W$. This means that $\mathcal{X}(G) = \mathcal{X}(G')$. □

It is worth noting that the graphs $G, G'$ of Theorem 2 can be distinguished by 2-WL (the 2-dimensional Weisfeiler-Leman algorithm), but not by 1-WL.

Proving that two graphs $G, G'$ have identical feature-matrix distributions $\mathcal{X}(G) = \mathcal{X}(G')$ as in Theorem 2 is the ultimate way of proving that they are not distinguishable by CRAWL. Yet, for more interesting graphs, we rarely have identical feature-matrix distributions. However, if the distributions are sufficiently close, we will still not be able to distinguish them. This will be the case for Theorem 1 (2). To quantify closeness, we use the *total variation distance* of the distributions. Recall that the total variation distance between two probability distributions $\mathcal{D}, \mathcal{D}'$ on the same finite sample space $\Omega$ is

$$\mathrm{dist}_{TV}(\mathcal{D}, \mathcal{D}') \coloneqq \max_{S \subseteq \Omega} |\Pr_{\mathcal{D}}(S) - \Pr_{\mathcal{D}'}(S)|.$$

It is known that the total variation distance is half the $\ell_1$-distance between the distributions, that is,

$$\mathrm{dist}_{TV}(\mathcal{D}, \mathcal{D}') = \frac{1}{2}\|\mathcal{D} - \mathcal{D}\|_1$$
$$= \frac{1}{2}\sum_{\omega \in \Omega} |\Pr_{\mathcal{D}}(\{\omega\}) - \Pr_{\mathcal{D}'}(\{\omega\})|.$$

Let $\varepsilon > 0$. We say that two graphs $G, G'$ are *$\varepsilon$-indistinguishable* by CRAWL with window size $s$, walk length $\ell$, and sample size $m$ if

$$\mathrm{dist}_{TV}(\mathcal{X}(G), \mathcal{X}(G')) < \frac{\varepsilon}{m}. \tag{1}$$

The rationale behind this definition is that if $\mathrm{dist}_{TV}(\mathcal{X}(G), \mathcal{X}(G')) < \frac{\varepsilon}{m}$ then for every property of feature matrices that CRAWL may want to use to distinguish the graphs, the expected numbers of samples with this property that CRAWL sees in both graphs are close together (assuming $\varepsilon$ is small).

Often, we want to make asymptotic statements, where we have two families of graphs $(G_n)_{n\geq1}$ and $(G'_n)_{n\geq1}$, typically of order $|G_n| = |G'_n| = \Theta(n)$, and classes $S, L, M$ of functions, such as the class $O(\log n)$ of logarithmic functions or the class $n^{O(1)}$ of polynomial functions. We say that $(G_n)_{n\geq1}$ and $(G'_n)_{n\geq1}$ are *indistinguishable* by CRAWL with window size $S$, walk length $L$, and sample size $M$ if for all $\varepsilon > 0$ and all $s \in S, \ell \in L, m \in M$ there is an $n$ such that $G_n, G'_n$ are $\varepsilon$-indistinguishable by CRAWL with window size $s(n)$, walk length $\ell(n)$, and sample size $m(n)$.

We could make similar definitions for distinguishability, but we omit them here, because there is an asymmetry between distinguishability and indistinguishability. To understand this, note that our notion of indistinguishability only looks at the features that are extracted on the random walks and not on how these features are processed by the 1D CNNs. Clearly, this ignores an important aspect of the CRAWL architecture. However, we would like to argue that for the results presented here this simplification is justified; if anything it makes the results stronger. Let us first look at the inexpressibility results (Theorem 1 (2) and Theorems 2 and 5). A prerequisite for CRAWL to distinguish two graphs is that the distributions of features observed on these

graphs are sufficiently different, so our notion of indistinguishability that just refers to the distributions of features yields a stronger notion of indistinguishability (indistinguishability in the feature-distribution sense implies indistinguishability in CRAWL). This means that our the indistinguishability results also apply to the actual CRAWL architecture.

For the distinguishability, the implication goes the other way and we need to be more careful. Our proof of Theorem 1 (1) (restated as Theorem 3 in the next section) does take this into account (see the discussion immediately after the statement of Theorem 3).

### 3.1 Proof of Theorem 1 (1)

Here is a precise quantitative version of the first part of Theorem 1.

**Theorem 3.** *For all $k \geq 1$ there are graphs $G_k, H_k$ of order $\mathcal{O}(k)$ such that CRAWL with window size $s = \mathcal{O}(k^2)$, walk length $\ell = \mathcal{O}(k^2)$, and sample size $m$ distinguishes $G_k$ and $H_k$ with probability at least $1 - \left(\frac{1}{2}\right)^m$ while $k$-WL (and hence $k$-dimensional GNNs) fails to distinguish $G_k$ and $H_k$.*

To prove the theorem we need to be careful with our theoretical model of distinguishability in CRAWL. It turns out that all we ever need to consider in the proof is the features observed within a single window of CRAWL. These features can be processed at once by the filter of the CNN. We assume that these filters can be arbitrary multilayer perceptrons (MLP) and hence share the universality property of MLPs that have at least one hidden layer: any continuous function on the compact domain $[0,1]^d$ (where $d$ is the number of all features in the feature matrix $X$ within a single window) can be approximated to arbitrary precision. While this is a strong assumption, it is common in expressivity results, e.g. (Xu et al., 2019).

In order to find pairs of graphs that $k$-WL is unable to distinguish, we do not need to know any details about the Weisfeiler-Leman algorithm (the interested reader is deferred to (Grohe, 2021; Kiefer, 2020)). Instead, we can use the following well-known inexpressibility result as a black box.

**Theorem 4** (Cai et al. (1992)). *For all $k \geq 1$ there are graphs $G_k, H_k$ such that $|G_k| = |H_k| = O(k)$, the graphs $G_k$ and $H_k$ are connected and 3-regular, and $k$-WL cannot distinguish $G_k$ and $H_k$.*

*Proof of Theorem 3.* We fix $k \geq 1$. Let $G_k, H_k$ be non-isomorphic graphs of order $O(k)$ that are connected, 3-regular, and not distinguishable by $k$-WL. Such graphs exist by Theorem 4.

Let $n := |G_k| = |H_k|$, and note that the number of edges of $G_k$ and $H_k$ is $(3/2)n$, because they are 3-regular. It is a well-known fact that the *cover time* of a connected graph of order $n$ with $m$ edges, that is, the expected time it takes a random walk starting from a random node to visit all nodes of the graph, is bounded from above by $4nm$ (Aleliunas et al., 1979). Thus, in our graphs $G_k, H_k$ the cover time is $6n^2$. Therefore, by Markov's inequality, a walk of length $12n^2$ visits all nodes with probability at least $1/2$.

We set the walk length $\ell$ and the window size $s$ to $12n^2$. Since $n$ is $O(k)$, this is $O(k^2)$.

Consider a single walk $W = (w_0, \ldots, w_\ell)$ of length $\ell$ in a graph $F \in \{G_k, H_k\}$, and assume that it visits all nodes, which happens with probability at least $1/2$ in either graph. Then the feature matrix $X = X(F, W)$, which we see at once in our window of size $s = \ell$, gives us a representation of the graph $F$. Of course, this representation depends on the specific order in which the walk traverses the graph. What this means is that if for two graphs $F, F'$ and walks $W, W'$ visiting all nodes of their respective graphs we have $X(F, W) = X(F', W')$, then the graphs $F, F'$ are isomorphic. This follows immediately from Observation 1. Note that the converse does not hold.

Since the above will rarely happen, we now define three sets of feature matrices that we may see:

$$R_G := \{X(G_k, W) \mid W \text{ walk on } G_k \text{ visiting every vertex}\},$$
$$R_H := \{X(H_k, W) \mid W \text{ walk on } H_k \text{ visiting every vertex}\},$$
$$S := \{X(F, W) \mid F \in \{G_k, H_k\}, W \text{ walk on } F \text{ not visiting every vertex}\}.$$

Observe that because we chose the walk length $\ell$ and the window size $s$ large enough we have:

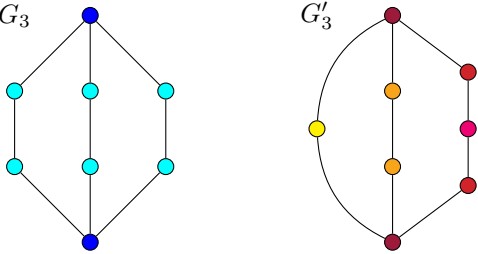

Figure 3: The graphs $G_3$ and $G_3'$ in the proof of Theorem 5 with their stable coloring computed by 1-WL

- for every walk on a graph $F \in \{G_k, H_k\}$, the feature matrix $X$ that we see belongs to one of the three sets, and with probability at least $1/2$ it belongs to $R_G$ or $R_H$;

- the three sets $R_G, R_H, S$ are mutually disjoint. For the non-isomorphic graphs $G_k, H_k$ we can never have $X(G_k, W) = X(H_k, W')$ such that a walk visiting all nodes will end up in the corresponding set $R_G$ or $R_H$. Through the equality features one can count the number of nodes visited in $W$, distinguishing $S$ from both $R_G$ and $R_H$.

Thus, if we sample a walk from $G_k$, with probability at least $1/2$ we see a feature matrix in $R_G$, and if we sample from $H_k$, we can never see a matrix in $R_G$. Let $f : [0,1]^{(\ell+1) \times (2s-1)} \to [0,1]$ be a continuous function that maps every feature matrix $X \in \{0,1\}^{(\ell+1) \times (2s-1)}$ to 1 if $X \in R_G$ and to 0 if $X \in R_H \cup S$.

Let us now run CRAWL on a graph $F \in \{G_k, H_k\}$. Since the window size $s$ equals the walk length $\ell$, CRAWL sees the full feature matrix $X(F, W)$ in a single window. Thus, by the universality of the CNN filters, it can approximate the function $f$ that checks whether a given feature matrix belongs to $R_G$ up to an additive error of $1/3$, allowing us to simply use $f$ as we can distinguish both cases independent of the error due to the approximation. We know that if we run CRAWL on $F = G_k$, with probability at least $1/2$ it holds that $X(F, W) \in R_G$ and thus $f(X(F, W)) = 1$. Otherwise, $f(X(F, W)) = 0$ with probability 1 as the three sets are disjoint. Hence, a CRAWL run based on a single random walk $W$ distinguishes the two graphs $G_k$ and $H_k$ with probability at least $1/2$. By exploiting that the error is one-sided and repeating the above argument $m$ times, one can distinguish the two graphs with probability $1 - (\frac{1}{2})^m$ using $m$ independent random walks. By simply relying on the universality of MLPs in this final step, the size of the CNN filter is exponential in $k$. This is not a limitation in this context, since $k$-WL also scales exponentially in $k$. However, in Appendix C we argue that with a more careful analysis the size of the MLP can actually be shown to have a polynomial upper bound. $\qquad \square$

We note that while the proof assumed that the random walk covered the whole graph and in addition the window size is chosen to encompass the complete walk, this is neither feasible nor necessary in practice. Instead, the theorem states that using a window size and path length of $O(k^2)$, CRAWL can detect differences in graphs which are inaccessible to $k$-GNNs. This is applicable to graphs of arbitrary size $n$ where the path of fixed size $O(k^2)$ (typically $k \leq 3$ as $k$-GNNs scale in $n^k$) is no longer able to cover the whole graph. In contrast to the theorem where a single walk (covering the whole graph) could suffice to tell apart two graphs, one needs multiple walks to distinguish the difference in the distributions of walks on real-world graphs as outlined earlier in this section.

## 3.2 Proof of Theorem 1 (2)

To prove the second part of the theorem, it will be necessary to briefly review the *1-dimensional Weisfeiler-Leman algorithm (1-WL)*, which is also known as *color refinement* and as *naive node classification*. The algorithm iteratively computes a partition of the nodes of its input graph. It is convenient to think of the classes of the partition as colors of the nodes. Initially, all nodes have the same color. Then in each iteration step, for all colors $c$ in the current coloring and all nodes $v, w$ of color $c$, the nodes $v$ and $w$ get different colors in the new coloring if there is some color $d$ such that $v$ and $w$ have different numbers of neighbors of

color $d$. This refinement process is repeated until the coloring is *stable*, that is, any two nodes $v, w$ of the same color $c$ have the same number of neighbors of any color $d$. We say that 1-WL *distinguishes* two graphs $G, G'$ if, after running the algorithm on the disjoint union $G \uplus G'$ of the two graphs, in the stable coloring of $G \uplus G'$ there is a color $c$ such that $G$ and $G'$ have a different number of nodes of color $c$.

For the results so far, it has not mattered if we allowed backtracking or not. Here, it makes a big difference. For the non-backtracking version, we obtain a stronger result with an easier proof. The following theorem is a precise quantitative statement of Theorem 1 (2).

**Theorem 5.** *There are families $(G_n)_{n \geq 1}$, $(G'_n)_{n \geq 1}$ of graphs of order $|G_n| = |G'_n| = 3n - 1$ with the following properties.*

> *(1) For all $n \geq 1$, 1-WL distinguishes $G_n$ and $G'_n$.*
>
> *(2) $(G_n)_{n \geq 1}$, $(G'_n)_{n \geq 1}$ are indistinguishable by the non-backtracking version of CRAWL with window size $s(n) = o(n)$ (regardless of the walk length and sample size).*
>
> *(3) $(G_n)_{n \geq 1}$ $(G'_n)_{n \geq 1}$ are indistinguishable by CRAWL with walk length $\ell(n) = O(n)$, and samples size $m(n) = n^{O(1)}$ (regardless of the window size).*

*Proof.* The graphs $G_n$ and $G'_n$ both consist of three internally disjoint paths with the same endnodes $x$ and $y$. In $G_n$ the lengths of all three paths is $n$. In $G'_n$, the length of the paths is $n-1, n, n+1$ (see Figure 3).

It is easy to see that 1-WL distinguishes the two graphs.

To prove assertion (2), let $s := 2n - 3$ be the window size. Then the length of the shortest cycle in $G_n$, $G'_n$ is $s + 2$. Now consider a non-backtracking walk $W = (w_1, \ldots, w_\ell)$ in either $G_n$ or $G'_n$ (of arbitrary length $\ell$). Intuitively, CRAWL only sees a simple path within its window size independent of which graph $W$ is sampled from and can thus not distinguish $G_n$ and $G'_n$. Formally, for all $i$ and $j$ with $i - s \leq j < i$ we have $w_i \neq w_j$, and unless $j = i - 1$, there is no edge between $w_i$ and $w_j$. Thus, $X(W) = X(W')$ for all walks $W'$ of the same length $\ell$, and since it does not matter which of the two graphs $G_n, G'_n$ the walks are from, it follows that $\mathcal{X}_{nb}(G_n) = \mathcal{X}_{nb}(G'_n)$.

To prove assertion (3) we use the fact that random walks of length $O(n)$ are very unlikely to traverse a path of length at least $n - 1$ from $x$ to $y$. It is well known that the expected traversal time is $\Theta(n^2)$ (this follows from the analysis of the gambler's ruin problem). However, this does not suffice for us. We need to bound the probability that a walk of length $O(n)$ is a traversal. Using a standard, Chernoff type tail bound, it is straightforward to prove that for every constant $c \geq 0$ there is a constant $d \geq 1$ such that the probability that a random walk of length $cn$ in either $G_n$ or $G'_n$ visits both $x$ and $y$ is at most $\exp(-n/d)$. As only walks visiting both $x$ and $y$ can differentiate between the two graphs, this gives us an upper bound of $\exp(-n/d)$ for the total variation distance between $\mathcal{X}(G_n)$ and $\mathcal{X}(G'_n)$. $\square$

Let us note that a bound on the walk length in assertion (3) of the previous theorem is necessary because the backtracking version of CRAWL with sufficiently long paths does seem to have the ability to distinguish the graphs $G_n, G'_n$ even with constant size windows. The intuitive argument is as follows: we first observe that, by going back and forth between a node and all its neighbors within its window, CRAWL can distinguish the two degree-3 nodes $x, y$ from the remaining degree-2 nodes. Thus, the feature matrix reflects traversal times between degree-3 nodes, and the distribution of traversal times is different in $G_n$ and $G'_n$. With sufficiently many samples, CRAWL can detect this. We leave it as future research to work out the quantitative details of this argument.

## 4 Experiments

We evaluate CRAWL on a range of standard graph learning benchmark datasets obtained from Dwivedi et al. (2020), Hu et al. (2020), and Dwivedi et al. (2022). In Appendix B we provide additional experimental results on the TUDataset (Morris et al., 2020a) and a direct comparison to Subgraph GNNs on the task of counting various substructures in synthetic graphs (Zhao et al., 2022).

### 4.1 Datasets

From the OGB project (Hu et al., 2020), we use the molecular property prediction dataset MOLPCBA with more than 400k molecules. Each of its 128 binary targets states whether a molecule is active towards a particular bioassay (a method that quantifies the effect of a substance on a particular kind of living cells or tissues). The dataset is adapted from MoleculeNet (Wu et al., 2018) and represents molecules as graphs of atoms. It contains multidimensional node and edge features which encode information such as atomic number and chirality. Additionally, it provides a train/val/test split that separates structurally different types of molecules for a more realistic experimental setting. On MOLPCBA, the performance is measured in terms of the average precision (AP). Further, we use four datasets from Dwivedi et al. (2020). The first dataset ZINC is a molecular regression dataset. It is a subset of 12k molecules from the larger ZINC database. The aim is to predict the *constrained solubility*, an important chemical property of molecules. The node label is the atomic number and the edge labels specify the bond type. The datasets CIFAR10 and MNIST are graph datasets derived from the corresponding image classification tasks and contain 60k and 70k graphs, respectively. The original images are modeled as networks of super-pixels. Both datasets are 10-class classification problems. The last dataset CSL is a synthetic dataset containing 150 *Cyclic Skip Link* graphs (Murphy et al., 2019). These are 4-regular graphs obtained by adding cords of a fixed length to a cycle. The formal definition and an example are provided in the appendix. The aim is to classify the graphs by their isomorphism class. Since all graphs are 4-regular and no node or edge features are provided, this task is unsolvable for standard MPGNNs.

We conduct additional experiments on the long-range graph benchmark (Dwivedi et al., 2022) which is a collection of datasets where capturing long-range interaction between vertices is crucial for performance. We use three long-range datasets: PASCALVOC-S is an image segmentation dataset where images are represented as graphs of superpixels. The task is a 21-class node classification problem. To apply CRAWL to this task we omit the pooling step and directly feed each vertex embedding into the classifier. PEPTIDES-FUNC and PEPTIDES-STRUCT model 16k peptides as molecular graphs. PEPTIDES-FUNC is a multi-class graph classification problem with 10 binary classes. Each class represents a biological function of the peptide. PEPTIDES-STRUCT aims to regress 11 molecular properties on the same set of graphs.

### 4.2 Experimental Setting

We adopt the training procedure specified by Dwivedi et al. (2020). In particular, the learning rate is initialized as $10^{-3}$ and decays with a factor of 0.5 if the performance on the validation set stagnates for 10 epochs. The training stops once the learning rate falls below $10^{-6}$. Dwivedi et al. (2020) also specify that networks need to stay within parameter budgets of either 100k or 500k parameters. This ensures a fairer comparison between different methods. For MNIST, CIFAR10, and CSL we train CRAWL models with the smaller budget of 100k since more baseline results are available in the literature. The OGB Project (Hu et al., 2020) does not specify a standardized training procedure or parameter budgets. For MOLPCBA, we train for 60 epochs and decay the learning rate once with a factor of 0.1 after epoch 50. On the long-range datasets PASCALVOC-SP, PEPTIDES-FUNC, and PEPTIDES-STRUCT we use the 500k parameter budget and found it helpful to switch to a cosine annealing schedule for the learning rate. For all datasets we use a walk length of $\ell = 50$ during training. For evaluation we increase this number to $\ell = 150$, except for MOLPCBA where we use $\ell = 100$ for efficiency. The window size $s$ was chosen to be 8 for all but the long-range datasets where we increased it to 16 to capture further dependencies.

All hyperparameters and the exact number of trainable parameters are listed in the appendix. We observed that our default setting of $\ell = 50$ and $s = 8$ to be a robust initial setting across all datasets. When we observed overfitting, decreasing the number of walks turned out to be a simple and effective measure to improve performance. In molecular datasets a window size of at least 6 is required to detect aromatic rings and in our ablation study in Section 5 we observe a clear performance drop on the molecular dataset ZINC when reducing $s$ below 6.

For each dataset we report the mean and standard deviation across several models trained with different random seeds. We follow the standardized procedure for each dataset and average over 10 models for MOLPCBA, 5 models for ZINC, CIFAR10, and MNIST and 4 models for PASCALVOC-SP, PEPTIDES-FUNC, and PEPTIDES-STRUCT. During inference, the output of each model depends on the sampled

Table 1: Performance achieved on ZINC, MNIST, CIFAR10, and MOLPCBA. A result marked with "†" indicates that the parameter budget was smaller than for CRAWL and "∗" marks results where no parameter budget was reported. We highlight the best results in **bold** and consider results with overlapping standard deviations statistically equivalent.

| | METHOD | ZINC TEST MAE | MNIST TEST ACC (%) | CIFAR10 TEST ACC (%) | MOLPCBA TEST AP |
|---|---|---|---|---|---|
| LEADERBOARD | GIN | $0.526 \pm 0.051$ | $96.485 \pm 0.252$ | $55.255 \pm 1.527$ | $0.2703 \pm 0.0023$ |
| | GRAPHSAGE | $0.398 \pm 0.002$ | $97.312 \pm 0.097$ | $65.767 \pm 0.308$ | - |
| | GAT | $0.384 \pm 0.007$ | $95.535 \pm 0.205$ | $64.223 \pm 0.455$ | - |
| | GCN | $0.367 \pm 0.011$ | $90.705 \pm 0.218$ | $55.710 \pm 0.381$ | $0.2483 \pm 0.0037$ |
| | 3WLGNN | $0.303 \pm 0.068$ | $95.075 \pm 0.961$ | $59.175 \pm 1.593$ | - |
| | GATEDGCN | $0.214 \pm 0.006$ | $97.340 \pm 0.143$ | $67.312 \pm 0.311$ | - |
| | PNA | $0.142 \pm 0.010$ | $\mathbf{97.940 \pm 0.120}$ | $70.350 \pm 0.630$ | $0.2838 \pm 0.0035$ |
| | GINE+ | - | - | - | $0.2917 \pm 0.0015$ |
| OTHER | DGN | $^{\dagger}0.168 \pm 0.003$ | - | $\mathbf{72.840 \pm 0.420}$ | - |
| | HIMP | $^{*}0.151 \pm 0.006$ | - | - | - |
| | GSN | $^{*}0.108 \pm 0.018$ | - | - | - |
| | SAT | $0.094 \pm 0.008$ | - | - | - |
| | SAN | $0.139 \pm 0.006$ | - | - | $0.2765 \pm 0.0042$ |
| | CIN | $\mathbf{0.079 \pm 0.006}$ | - | - | - |
| | NESTED GIN | - | - | - | $0.2832 \pm 0.0041$ |
| | GIN-AK+ | $\mathbf{0.080 \pm 0.001}$ | - | $\mathbf{72.190 \pm 0.130}$ | $0.2930 \pm 0.0044$ |
| OUR | CRAWL | $\mathbf{0.085 \pm 0.004}$ | $\mathbf{97.944 \pm 0.050}$ | $69.013 \pm 0.259$ | $\mathbf{0.2986 \pm 0.0025}$ |

random walks. Thus, each model's performance is given by the average performance over 10 evaluations based on different random walks. This internal model deviation, that is, the impact of the random walks on the performance, is substantially lower than the differences between models. We thus focus on the mean and standard deviation between CRAWL models when comparing to other methods. In the appendix we provide extended results that additionally specify the internal model deviation.

## 4.3 Baselines

We compare the results obtained with CRAWL to a wide range of graph learning methods. Our main baselines are numerous message passing GNN and Graph Transformer architectures that have been proposed in recent years (see Section 1.1). We report values as provided in the literature and official leaderboards[3]. For a fair and direct comparison, we exclude results of ensemble methods and models pre-trained on additional data.

## 4.4 Results

Table 1 provides our results on ZINC, MNIST, CIFAR10, and MOLPCBA. On the ZINC dataset, CRAWL achieves an MAE of 0.085. This is approximately a 40% improvement over the current best standard MPGNN (PNA) on ZINC. The result is on par (within standard deviation) of Cellular GNNs (CIN) (Bodnar et al., 2021) and "GNN as Kernel" (GIN-AK+) (Zhao et al., 2022), which are state of the art on ZINC. CRAWL's performance on the MNIST dataset is on par with PNA (within standard deviation), which is also the state of the art on this dataset. On CIFAR10, CRAWL achieves the fourth-highest accuracy among the compared approaches. On MOLPCBA, CRAWL yields state-of-the-art results and improves upon all compared architectures. We note that the baselines include architectures strictly stronger than 1-WL, such as Nested GIN (Zhang & Li, 2021). CRAWL yields better results, indicating that the WL-incomparability does not keep our method from outperforming higher-order MPGNNs.

---

[3]Leaderboards: https://ogb.stanford.edu/docs/leader_graphprop and
https://github.com/graphdeeplearning/benchmarking-gnns/blob/master-may2022/docs/07_leaderboards.md

Table 2: Performance on the PASCALVOC-SP, PEPTIDES-FUNC, and PEPTIDES-STRUCT datasets from the Long-Range Graph Benchmark. Baseline results are reported by Dwivedi et al. (2022)

| METHOD | PASCALVOC-SP TEST F1 | PEPTIDES-FUNC TEST AP | PEPTIDES-STRUCT TEST MAE |
|---|---|---|---|
| GCN | $0.1268 \pm 0.0060$ | $0.5930 \pm 0.0023$ | $0.3496 \pm 0.0013$ |
| GINE | $0.1265 \pm 0.0076$ | $0.5498 \pm 0.0079$ | $0.3547 \pm 0.0045$ |
| GATEDGCN | $0.2873 \pm 0.0219$ | $0.6069 \pm 0.0035$ | $0.3357 \pm 0.0006$ |
| TRANSFORMER | $0.2694 \pm 0.0098$ | $0.6326 \pm 0.0126$ | $\mathbf{0.2529 \pm 0.0016}$ |
| SAN | $0.3230 \pm 0.0039$ | $0.6439 \pm 0.0075$ | $0.2545 \pm 0.0012$ |
| CRAWL | $\mathbf{0.4588 \pm 0.0079}$ | $\mathbf{0.7074 \pm 0.0032}$ | $\mathbf{0.2506 \pm 0.0022}$ |

Table 3: Test accuracy achieved on CSL with and without Laplacian eigenvectors (Dwivedi et al., 2020). As these node features already encode the solution, unsurprisingly most models perform well.

| METHOD | CSL TEST ACC (%) | CSL+LAP TEST ACC (%) |
|---|---|---|
| GIN | $\leq 10.0$ | $99.333 \pm 1.333$ |
| GRAPHSAGE | $\leq 10.0$ | $99.933 \pm 0.467$ |
| GAT | $\leq 10.0$ | $99.933 \pm 0.467$ |
| GCN | $\leq 10.0$ | $\mathbf{100.000 \pm 0.000}$ |
| 3WLGNN | $95.700 \pm 14.850$ | $30.533 \pm 9.863$ |
| GATEDGCN | $\leq 10.0$ | $99.600 \pm 1.083$ |
| CRAWL | $\mathbf{100.000 \pm 0.000}$ | $\mathbf{100.000 \pm 0.000}$ |

Table 2 provides the results obtained for the long-range graph benchmark datasets. CRAWL achieves state-of-the-art performance on all three datasets. On PASCALVOC-SP and PEPTIDES-FUNC our architecture is able to improve the previous best result by 13 and 6 percentage points respectively. The margin on PEPTIDES-STRUCT is less significant, but CRAWL is still able to yield the lowest MAE compared to both GNNs and Graph Transformers. These results validate empirically that CRAWL is able to capture long-range interaction between nodes through the use of random walks.

The results on CSL are reported in Table 3. We consider two variants of CSL, the pure task and an easier variant in which node features based on Laplacian eigenvectors are added as suggested by Dwivedi et al. (2020). Already on the pure task, CRAWL achieves an accuracy of 100%, none of the 5 CRAWL models misclassified a single graph in the test folds. 3WLGNN is also theoretically capable of solving the pure task and achieves almost 100% accuracy, while MPGNNs cannot distinguish the pure 4-regular CSL graphs and achieve at most 10% accuracy. With Laplacian features that essentially encode the solution, all approaches (except 3WLGNN) achieve results close to 100%. Thus, the CSL dataset showcases the high theoretical expressivity of CRAWL. High expressivity also helps in subgraph counting which is effectively solved by both CRAWL and subgraph GNNs such as GIN-AK+ as shown in Appendix B.4. In contrast, pure MPGNNs are theoretically and practically unable to count anything more complicated than stars and even counting those seems to be challenging in practice.

Overall, CRAWL performs very well on a variety of datasets across several domains.

## 5 Ablation Study

We perform an ablation study to understand how the key aspects of CRAWL influence the empirical performance. We aim to answer two main questions:

- How useful are the identity and adjacency features we construct for the walks?

Table 4: Results of our ablation study. Node features $F^V$, edge features $F^E$, adjacency encoding $A$, and identity encoding $I$. Walk strategies no-backtrack (NB) and uniform (UN).

| Features | Walks | ZINC (MAE) | MOLPCBA (AP) | CSL (Acc) |
|---|---|---|---|---|
| $F^V+F^E$ | UN | $0.19768 \pm 0.01159$ | $0.28364 \pm 0.00201$ | $0.06000 \pm 0.04422$ |
| $F^V+F^E$ | NB | $0.15475 \pm 0.00350$ | $0.29613 \pm 0.00209$ | $0.06000 \pm 0.04422$ |
| $F^V+F^E+A$ | UN | $0.10039 \pm 0.00514$ | - | $0.97467 \pm 0.02587$ |
| $F^V+F^E+A$ | NB | $0.08656 \pm 0.00310$ | - | $0.99933 \pm 0.00133$ |
| $F^V+F^E+I$ | UN | $0.10940 \pm 0.00698$ | - | $0.70733 \pm 0.07658$ |
| $F^V+F^E+I$ | NB | $0.09345 \pm 0.00219$ | - | $0.97133 \pm 0.00859$ |
| $F^V+F^E+I+A$ | UN | $0.09368 \pm 0.00232$ | $0.28522 \pm 0.00317$ | $0.96267 \pm 0.02037$ |
| $F^V+F^E+I+A$ | NB | $0.08456 \pm 0.00352$ | $0.29863 \pm 0.00249$ | $1.00000 \pm 0.00000$ |

- How do different strategies for sampling random walks impact the performance?

- How do window size $s$ and the number of walks $m$ influence the performance?

Here, we use the ZINC, MOLPCBA, and CSL datasets to answer these questions empirically. We trained multiple versions of CRAWL with varying amounts of structural features used in the walk feature matrices. The simplest version only uses the sequences of node and edge features without any structural information. For ZINC and CSL, we also train intermediate versions using either the identity or the adjacency encoding, but not both. We omit these for MOLPCBA to save computational resources. Finally, we measure the performance of the standard CRAWL architecture, where both encodings are incorporated into the walk feature matrices. For each version, we compute the performance with both walk strategies.

On each dataset, the experimental setup and hyperparameters are identical to those used in the previous experiments on both datasets. In particular, we train five models with different seeds and provide the average performance as well as the standard deviation across models. Note that we repeat the experiment independently for each walk strategy. Switching walk strategies between training and evaluation does not yield good results.

Table 4 reports the performance of each studied version of CRAWL. On ZINC, the networks without any structural encoding yield the worst predictions. Adding either the adjacency or the identity encoding improves the results substantially. The best results are obtained when both encodings are utilized and non-backtracking walks are used. On MOLPCBA, the best performance is also obtained with full structural encodings. However, the improvement over the version without the encodings is only marginal. Again, non-backtracking walks perform significantly better than uniform walks. On CSL, the only version to achieve a perfect accuracy of 100% is the one with all structural encodings and non-backtracking walks. Note that the version without any encodings can only guess on CSL since this dataset has no node features (we are not using the Laplacian features here).

Overall, the structural encodings of the walk feature matrices yield a measurable performance increase on all three datasets. However, the margin of the improvement varies significantly and depends on the specific dataset. For some tasks such as MOLPCBA, CRAWL yields highly competitive results even when only the sequences of node and edge features are considered in the walk feature matrices.

Finally, the non-backtracking walks consistently outperform the uniform walks. This could be attributed to their ability to traverse sparse substructures quickly. On sparse graphs with limited degree such as molecules, uniform walks will backtrack often. Thus, the subgraphs induced by windows of a fixed size $s$ on uniform walks tend to be smaller than for non-backtracking walks. This limits the method's power to evaluate local substructures and long-range dependencies. On all three datasets used in this ablation study, these effects seem to cause a significant loss in performance.

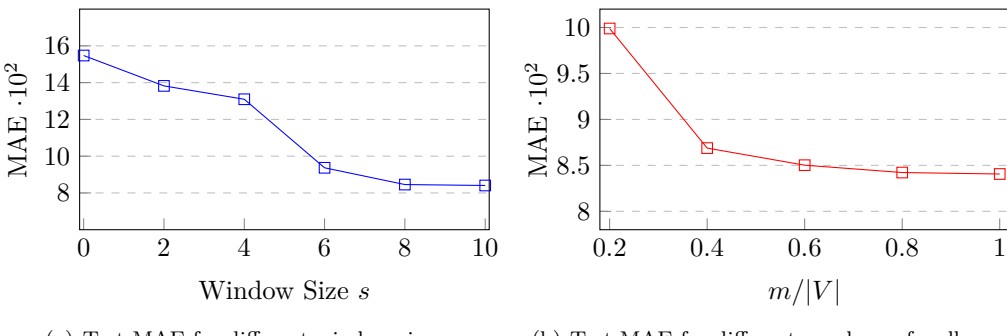

(a) Test MAE for different window sizes $s$.  (b) Test MAE for different numbers of walks $m$.

Figure 4: Ablation on the ZINC dataset. We study the influence of $s$ and $m$ on the test MAE.

### 5.1 Influence of $s$ and $m$

Let us further study how the values of the window size $s$ and the number of sampled walks $m$ influence the empirical performance of CRAWL. Generally, we expect performance to improve as both parameters increase: Larger window sizes $s$ allow the model to view larger structural features as the expected size of the induced subgraph increases, while a larger number of walks ensures a more reliable traversal of important features. To test this hypothesis, we conduct an additional ablation study on the ZINC dataset.

First, we evaluate the effect of $s$ on the model performance. To this end, we train models with varying window sizes of $s \in \{0, 2, 4, 6, 8, 10\}$ while all other parameters remain as in our main experiment where we used $s = 8$. We train 3 models for each $s$ and provide the mean test MAE as a function of $s$ in Figure 4(a). We observe that the performance improves monotonically as $s$ increases. The decrease in MAE is especially significant between $s = 4$ and $s = 6$. This jump can be explained by the importance of benzene rings in organic molecules and in the ZINC dataset in particular. A window of size $s < 6$ is insufficient to detect these structures, which explains the strong improvement around this threshold. The performance saturates around $s = 8$ and is not improved further for $s = 10$. Secondly, we aim to quantify the effect of $m$. We use the trained models from our main experiment and evaluate them on the test split of ZINC with different numbers of walks $m$. We vary the number of walks $m = p \cdot |V|$ where $p \in \{0.2, 0.4, 0.6, 0.8, 1.0\}$ and $|V|$ is the number of vertices in a given test graph. All other parameters are unchanged. Figure 4(b) provides the mean test MAE as a function of $p$. The error decreases monotonically for larger values of $m$, as predicted. The improvement from $m = 0.8|V|$ to $m = |V|$ is marginal. We note that the walk length $\ell$ has a similar effect on the performance as the product $m \cdot \ell$ determines how densely a graph is covered in walks. In this ablation study the walk length is fixed at $\ell = 150$ during inference, as in our main experiment.

Overall, we conclude that our hypothesis seems to hold as larger values for $s$ and $m$ seem to improve performance.

## 6 Conclusion

We have introduced a novel neural network architecture CRAWL for graph learning that is based on random walks and 1D CNNs. We demonstrated the effectiveness of this approach on a variety of graph level tasks on which it is able to outperform or at least match state-of-the-art GNNs. On a theoretical level, the expressiveness of CRAWL is incomparable to the Weisfeiler-Leman hierarchy which bounds standard message passing GNNs. By construction, CRAWL can detect arbitrary substructures up to the size of its local window. Thus CRAWL is able to extract useful features from highly symmetrical graphs such as the 4-regular graphs of CSL on which pure MPGNNs fail due to the lack of expressiveness. This way random walks allow CRAWL to escape the Weisfeiler-Leman hierarchy and solve tasks that are impossible for MPGNNs.

CRAWL can be viewed as an attempt to process random walks and the structures they induce with end-to-end neural networks. The strong empirical performance demonstrates the potential of this general approach.

However, many variations remain to be explored, including different walk strategies, variations in the walk features, and alternative pooling functions for pooling walklet embeddings into nodes or edges. Since our CRAWL layer is compatible with MPGNNs and also with Graph Transformers, it raises the question of what would happen if we combine all three techniques as all three methods focus on different aspects of the graph structure. Through a unified architecture that for example runs all three aggregation methods in parallel we would thus anticipate a performance gain due to synergies. Beyond plain 1D-CNNs, other deep learning architectures for sequential data, such as LSTMs or sequence transformers, could be used to process random walks. Furthermore, extending the experimental framework to node-level tasks and motif counting would open new application areas for walk-based approaches. In both cases, one needs to scale CRAWL to work on individual large graphs instead of many medium-sized ones. Next to the combination of the full CRAWL layer with other techniques, its components such as the adjacency encoding could help improving other existing models based on processing subgraphs or random walks.

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

# A  Model Details

In this appendix, we provide additional definitions and details on CRAWL, including the random walks that we sample and the exact architecture of the CNN in each layer.

## A.1  Random Walks

A walk of length $\ell \in \mathbb{N}$ in a graph $G = (V, E)$ is a sequence of nodes $(v_0, \ldots, v_\ell) \in V^{\ell+1}$ with $v_{i-1} v_i \in E$ for all $i \in [\ell]$. A random walk in a graph is obtained by starting at some initial node $v_0 \in V$ and then iteratively sampling the next node $v_{i+1}$ randomly from the neighbors $N_G(v_i)$ of the current node $v_i$. We consider two different random walk strategies: *uniform* and *non-backtracking*. The uniform walks are obtained by sampling the next node uniformly from all neighbors:

$$v_{i+1} \sim \mathcal{U}\big(N_G(v_i)\big).$$

On sparse graphs with nodes of small degree (such as molecules) this walk strategy has a tendency to backtrack often. This slows the traversal of the graph and interferes with the discovery of long-range patterns. The non-backtracking walk strategy addresses this issue by excluding the previous node from the sampling (unless the degree is one):

$$v_{i+1} \sim \mathcal{D}_{\mathrm{NB}}(v_i) \quad \text{with} \quad \mathcal{D}_{\mathrm{NB}}(v_i) = \begin{cases} \mathcal{U}\big(N_G(v_i)\big), & \text{if } i = 0 \vee \deg(v_i) = 1 \\ \mathcal{U}\big(N_G(v_i) \backslash \{v_{i-1}\}\big), & \text{else.} \end{cases}$$

The choice of the walk strategy is a hyperparameter of CRAWL. In our experiments the non-backtracking strategy usually performs better as shown in Section 5.

## A.2  Convolution Module

Here, we describe the architecture used for the 1D CNN network $\mathrm{CNN}^t$ in each layer $t$. Let $\mathrm{Conv1D}(d, d', k)$ be a standard 1D convolution with input feature dimension $d$, output feature dimension $d'$, kernel size $k$ and no bias. This module has $d \cdot d' \cdot k$ trainable parameters. The term scales poorly for larger hidden dimensions $d$, since the square of this dimension is scaled with an additional factor of $k$, which we typically set to 9 or more.

To address this issue we leverage *Depthwise Separable Convolutions*, as suggested by Chollet (2017). This method is most commonly applied to 2D data in Computer Vision, but it can also be utilized for 1D convolutions. It decomposes one convolution with kernel size $k$ into two convolutions: The first convolution is a standard 1D convolution with kernel size 1. The second convolution is a depthwise convolution with kernel size $k$, which convolves each channel individually and therefore only requires $k \cdot d'$ parameters. The second convolution is succeeded by a Batch Norm layer and a ReLU activation function. Note that there is no non-linearity between the two convolutions. These operations effectively simulate a standard convolution with kernel size $k$ but require substantially less memory and runtime.

After the ReLU activation, we apply an additional (standard) convolution with kernel size 1, followed by another ReLU non-linearity. This final convolution increases the expressiveness of our convolution module which could otherwise only learn linearly separable functions. This would limit its ability to distinguish the binary patterns that encode identity and adjacency.

The full stack of operations effectively applies a 2-layer MLP to each sliding window position of the walk feature tensor. Overall, $\mathrm{CNN}^t$ is composed of the following operations:
$$\mathrm{Conv1D}(d, d', 1) \rightarrow \mathrm{Conv1D}^{dw}(d', d', k) \rightarrow \mathrm{BatchNorm} \rightarrow \mathrm{ReLU} \rightarrow \mathrm{Conv1D}(d', d', 1) \rightarrow \mathrm{ReLU}$$

Here, Conv1D is a standard 1D convolution and $\mathrm{Conv1D}^{dw}$ is a depthwise convolution. The total number of parameters of one such module (without the affine transformation of the Batch Norm) is equal to $dd' + kd + d'^2$.

Table 5: Hyperparameters used in each experiment.

| | ZINC | CIFAR10 | MNIST | CSL | MOLPCBA | PASCAL VOC-SP | PEPTIDES -FUNC | PEPTIDES -STRUCT |
|---|---|---|---|---|---|---|---|---|
| $L$ | 3 | 3 | 3 | 2 | 5 | 6 | 6 | 6 |
| $d$ | 147 | 75 | 75 | 90 | 400 | 100 | 100 | 100 |
| $s$ | 8 | 8 | 8 | 8 | 8 | 16 | 16 | 16 |
| pool | sum | mean | mean | mean | mean | sum | sum | sum |
| out | mlp | mlp | mlp | mlp | linear | mlp | mlp | mlp |
| $\ell_{\text{train}}$ | 50 | 50 | 50 | 50 | 50 | 50 | 50 | 50 |
| $\ell_{\text{eval}}$ | 150 | 150 | 150 | 150 | 100 | 150 | 150 | 150 |
| $p*$ | 1 | 1 | 1 | 1 | 0.2 | 0.2 | 0.2 | 0.2 |
| walk strat. | nb | nb | nb | nb | nb | nb | nb | nb |
| $r_{\text{val}}$ | 2 | 2 | 2 | 5 | 2 | 2 | 2 | 2 |
| $r_{\text{test}}$ | 10 | 10 | 10 | 10 | 10 | 5 | 5 | 5 |
| dropout | 0.0 | 0.0 | 0.0 | 0.0 | 0.25 | 0.1 | 0.1 | 0.2 |
| learning rate | 0.001 | 0.001 | 0.001 | 0.001 | 0.001 | 0.001 | 0.001 | 0.001 |
| patience | 10 | 10 | 10 | 20 | - | - | - | - |
| epochs | - | - | - | - | 60 | 300 | 500 | 500 |
| batch size | 50 | 50 | 50 | 50 | 100 | 50 | 50 | 50 |
| virtual node | Yes | No | No | No | Yes | Yes | Yes | Yes |
| #Params | 497,743 | 109,660 | 109,360 | 104,140 | 6,115,728 | 495,021 | 510,910 | 511,011 |

## A.3 Architecture

The architecture we use in the experiments works as follows. We then stack multiple CRAWL layers with residual connections. In each CRAWL layer, we typically choose $s = 8$. The update MLP $U^{(t)}$ has a single hidden layer of dimension $2d$ with ReLU activation and a linear output layer with $d$ units. After the final CRAWL layer, we apply batch normalization and a ReLU activation to the latent node embeddings before we perform a global pooling step. As pooling we use either sum-pooling or mean-pooling. Finally, a simple feedforward neural network is used to produce a graph-level output which can then be used in classification and regression tasks. In our experiments, we use either an MLP with one hidden layer of dimension $d$ or a single linear layer.

Since CRAWL layers are based on iteratively updating latent node embeddings, they are fully compatible with conventional message passing layers and related techniques such as virtual nodes (Gilmer et al., 2017; Li et al., 2017; Ishiguro et al., 2019). In our experiments, we use virtual nodes whenever this increases validation performance. A detailed explanation of our virtual node layer is provided in Appendix A.5. Combining CRAWL with message passing layers is left as future work.

## A.4 Hyperparameters

Table 5 provides the hyperparameters used in each experiment. The number of layers $L$ and dimension $d$ were tuned to meet the parameter budgets for each dataset. The local window size $s$ is set to 8 by default and increased to 16 on the long-range datasets. As global pooling function we use either mean- our sum-pooling. The architecture of the final output network is either a 2-layer MLP or a simple linear transformation. We denote the repetitions with different random walks to estimate the IMD in Section B.1 with $r_{\text{val}}$ and $r_{\text{test}}$.

Hyperparameters for the walks and the training procedure: The number of random walk steps during training $\ell_{\text{train}}$ is set to 50 and increased to $\ell_{\text{eval}}=150$ during evaluation. The probability of starting a walk from each node during training $p*$ is chosen as 1 by default. On MOLPCBA and the long-range datasets we set $p* = 0.2$ to reduce overfitting. The walk strategy (either uniform (un) or non-backtracking (nb))

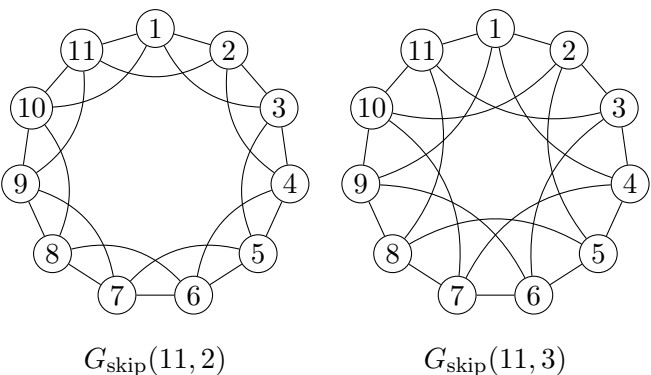

$$G_{\text{skip}}(11, 2) \qquad\qquad G_{\text{skip}}(11, 3)$$

Figure 5: Two cyclic skip-link graphs (see Murphy et al., 2019) with 11 nodes and a skip distance of 2 and 3 respectively.

## A.5 Virtual Node

Gilmer et al. (2017), Li et al. (2017), and Ishiguro et al. (2019) suggested the use of a *virtual node* to enhance GNNs for chemical datasets. Intuitively, a special node is inserted into the graph that is connected to all other nodes. This node aggregates the states of all other nodes and uses this information to update its own state. The virtual node has its own distinct update function which is not shared by other nodes. The updated state is then sent back to all nodes in the graph. Effectively, a virtual node allows global information flow after each layer.

Formally, a virtual node updates a latent state $h_{vn}^t \in \mathbb{R}^d$, where $h_{vn}^t$ is computed after the $t$-th layer and $h_{vn}^0$ is initialized as a zero vector. The update procedure is defined by:

$$h_{vn}^t = U_{vn}^t \left( h_{vn}^{t-1} + \sum_{v \in V} h^t(v) \right)$$
$$\tilde{h}^t(v) = h^t(v) + h_{vn}^t.$$

Here, $U_{vn}^t$ is a trainable MLP and $h^t$ is the latent node embedding computed by the $t$-th CRAWL layer. $\tilde{h}^t$ is an updated node embedding that is used as the input for the next CRAWL layer instead of $h^t$. In our experiments, we choose $U_{vn}^t$ to contain a single hidden layer of dimension $d$. When using a virtual node, we perform this update step after every CRAWL layer, except for the last one.

Note that we view the virtual node as an intermediate update step that is placed between our CRAWL layers to allow for global communication between nodes. No additional node is actually added to the graph and, most importantly, the "virtual node" does not occur in the random walks sampled by CRAWL.

## A.6 Cross Validation on CSL

Let us briefly discuss the experimental protocol used for the CSL dataset. Unlike the other benchmark datasets provided by Dwivedi et al. (2020), CSL is evaluated with 5-fold cross-validation. We use the 5-fold split Dwivedi et al. (2020) provide in their repository. In each training run, three folds are used for training and one is used for validation and model selection. After training, the remaining fold is used for testing.

Finally, Figure 5 provides an example of two skip-link graphs. The task of CSL is to classify such graphs by their isomorphism class.

Table 6: Statistics for our main datasets.

|  | #Graphs | Avg. $|V|$ | Avg. $|E|$ | #Classes/#Tasks |
|---|---|---|---|---|
| ZINC | 10000 / 1000 / 1000 | 23.1 | 49.8 | 1 |
| CIFAR10 | 45000 / 5000 / 10000 | 117.6 | 1129.8 | 10 |
| MNIST | 55000 / 5000 / 10000 | 80.98 | 785.39 | 10 |
| CSL | 150 | 41 | 82 | 10 |
| MOLPCBA | 350343 / 43793 / 43793 | 25.6 | 55.4 | 128 |
| PASCALVOC-SP | 8498/1428/1429 | 479.40 | 2710.48 | 21 |
| PEPTIDES-FUNC | 10873/2331/2331 | 150.94 | 307.30 | 10 |
| PEPTIDES-STRUCT | 10873/2331/2331 | 150.94 | 307.30 | 11 |

Table 7: Extended results for CRAWL on all datasets. Note that different metrics are used to measure the performance on the datasets. For each experiment we provide the cross model deviation (CMD) and the internal model deviation (IMD).

| DATASET / MODEL | METRIC | TEST | | | VALIDATION | | | TRAIN | |
|---|---|---|---|---|---|---|---|---|---|
| | | SCORE | CMD | IMD | SCORE | CMD | IMD | SCORE | CMD |
| ZINC | MAE | 0.08456 | ± 0.00352 | ± 0.00116 | 0.11398 | ± 0.00447 | ± 0.00121 | 0.04913 | ± 0.00887 |
| CIFAR10 | Acc. | 0.69013 | ± 0.00259 | ± 0.00158 | 0.70052 | ± 0.00307 | ± 0.00060 | 0.79180 | ± 0.01956 |
| MNIST | Acc. | 0.97944 | ± 0.00050 | ± 0.00055 | 0.98106 | ± 0.00110 | ± 0.00030 | 0.99044 | ± 0.00090 |
| CSL | Acc. | 1.00000 | ± 0.00000 | ± 0.00000 | 1.00000 | ± 0.00000 | ± 0.00000 | 1.00000 | ± 0.00000 |
| MOLPCBA | AP | 0.29863 | ± 0.00249 | ± 0.00055 | 0.30746 | ± 0.00195 | ± 0.00027 | 0.54889 | ± 0.01021 |
| PESCALVOC-SP | F1 | 0.45878 | ± 0.00794 | ± 0.00076 | 0.45326 | ± 0.00356 | ± 0.00031 | 0.91138 | ± 0.00361 |
| PEPTIDES-FUNC | AP | 0.70735 | ± 0.00317 | ± 0.00059 | 0.72716 | ± 0.00272 | ± 0.00064 | 0.98034 | ± 0.00771 |
| PEPTIDES-STRUCT | MAE | 0.25057 | ± 0.00224 | ± 0.00019 | 0.24089 | ± 0.00150 | ± 0.00016 | 0.20705 | ± 0.00870 |

## B   Extended Results

The dataset statistics of all datasets used in the main paper are given in Table 6. In the following, we provide experimental results in full detail.

### B.1   Detailed Results

Table 7 provides the full results from our experimental evaluation. It reports the performance on the train, validation, and test data.

Recall that the output of CRAWL is a random variable. The predictions for a given input graph may vary when different random walks are sampled. To quantify this additional source of randomness, we measure two deviations for each experiment: The cross model deviation (CMD) and the internal model deviation (IMD). For clarity, let us define these terms formally. For each experiment, we perform $q \in \mathbb{N}$ training runs with different random seeds. Let $m_i$ be the model obtained in the $i$-th training run with $i \in [q]$. When evaluating (both on test and validation data), we evaluate each model $r \in \mathbb{N}$ times, with different random walks in each evaluation run. Let $p_{i,j} \in \mathbb{R}$ measure the performance achieved by the model $m_i$ in its $j$-th evaluation run. Note that the unit of $p_{i,j}$ varies between experiments (Accuracy, MAE, ...). We formally define the *internal model deviation* as

$$\text{IMD} = \frac{1}{q} \cdot \sum_{1 \leq i \leq q} \text{STD}\left(\{p_{i,j} \mid 1 \leq j \leq r\}\right),$$

where $\text{STD}(\cdot)$ is the standard deviation of a given distribution. Intuitively, the IMD measures how much the performance of a trained model varies when applying it multiple times to the same input. It quantifies how the model performance depends on the random walks that are sampled during evaluation.

We formally define the *cross model deviation* as

$$\text{CMD} = \text{STD}\left(\left\{\frac{1}{r} \cdot \sum_{1 \leq j \leq r} p_{i,j} \mid 1 \leq i \leq q\right\}\right).$$

Table 8: Average time per epoch for CRAWL. The reported times are averaged over a training run and include the time used to perform a validation run after each training epoch. We also provide reported training times for several baselines as reported in the literature (Dwivedi et al., 2020; 2022; Zhao et al., 2022). Note that absolute training times are sparsely reported and depend on implementation details and hardware. We report these values as a rough guideline of how long the training of standard MPGNNs and stronger architectures usually takes.

| DATASET | CRAWL | GCN | GATEDGCN | TRANSFORMER | SAN | GIN-AK+ |
|---|---|---|---|---|---|---|
| ZINC | 10.0s | 12.8s | 10.7s | 27.78s | 106.0s | 9.4s |
| CIFAR10 | 265.0 | 109.7s | 154.2s | - | - | 241.1 |
| MOLPCBA | 458.4s | - | - | - | 883.0s | - |
| PASCALVOC-SP | 47.8s | 8.8s | 12.0s | 13.0s | 179.0s | - |
| PEPTIDES-FUNC | 18.5s | 3.0s | 3.3s | 5.8s | 49.1s | - |
| PEPTIDES-STRUCT | 18.4s | 2.6s | 3.3s | 5.9s | 49.7s | - |

Table 9: Accuracy on Social Datasets.

| Method | | COLLAB Test Acc | IMDB-MULTI Test Acc | REDDIT-BIN Test Acc |
|---|---|---|---|---|
| WL-Kernel | (Shervashidze et al., 2011) | $78.9 \pm 1.9$ | $50.9 \pm 3.8$ | $81.0 \pm 3.1$ |
| WEGL | (Kolouri et al., 2021) | $79.8 \pm 1.5$ | $52.0 \pm 4.1$ | $92.0 \pm 0.8$ |
| GNTK | (Du et al., 2019) | $83.6 \pm 1.0$ | $52.8 \pm 4.6$ | - |
| DGCNN | (Zhang et al., 2018) | $73.8 \pm 0.5$ | $47.8 \pm 0.9$ | - |
| 3WLGNN | (Maron et al., 2019a) | $80.7 \pm 1.7$ | $50.5 \pm 3.6$ | - |
| GIN | (Xu et al., 2019) | $80.2 \pm 1.9$ | $52.3 \pm 2.8$ | $92.4 \pm 2.5$ |
| GSN | (Bouritsas et al., 2020) | $\mathbf{85.5 \pm 1.2}$ | $\mathbf{54.3 \pm 3.3}$ | - |
| CRAWL | | $80.40\% \pm 1.50$ | $47.77\% \pm 3.87$ | $\mathbf{92.75\% \pm 2.16}$ |

The CMD measures the deviation of the average model performance between different training runs. It therefore quantifies how the model performance depends on the random initialization of the network parameters before training.

In the main section, we only reported the CMD for simplicity. Note that the CMD is significantly larger then the IMD across all experiments. Therefore, trained CRAWL models can reliably produce high quality predictions, despite their dependence on randomly sampled walks.

## B.2 Runtime

Table 8 provides runtime per epoch observed during training. All experiments were run on a machine with 64GB RAM, an Intel Xeon 8160 CPU and an Nvidia Tesla V100 GPU with 16GB GPU memory. From the table we observe that the runtime of CRAWL does not differ much from common MPGNNs. While simpler architectures like GCN can run faster on larger datasets, CRAWL has no runtime disadvantage when compared to more advanced architectures like Graph Transformers (SAN) or subgraph GNNs (GIN-AK+).

## B.3 Additional Experiments

In this section we evaluate CRAWL on commonly used benchmark datasets from the domain of social networks. We use a subset from the TUDataset (Morris et al., 2020a), a list of typically small graph datasets from different domains e.g. chemistry, bioinformatics, and social networks. We focus on three datasets originally proposed by Yanardag & Vishwanathan (2015): COLLAB, a scientific collaboration dataset, IMDB-MULTI, a multiclass dataset of movie collaboration of actors/actresses, and REDDIT-BIN, a balanced binary classification dataset of Reddit users which discussed together in a thread. These datasets do not have any node or edge features and the tasks have to be solved purely with the structure of the graphs.

Table 10: Performance on the subgraph counting task.

| Method | Triangle | Tailed Triangle | 3-Star | 4-Cycle |
|---|---|---|---|---|
| GCN | 0.4186 | 0.3248 | 0.1798 | 0.2822 |
| GIN | 0.3569 | 0.2373 | 0.0224 | 0.2185 |
| PNA | 0.3532 | 0.2648 | 0.1278 | 0.2430 |
| GCN-AK+ | 0.0137 | 0.0134 | 0.0174 | 0.0174 |
| GIN-AK+ | 0.0137 | **0.0112** | 0.0150 | 0.0150 |
| PNA-AK+ | **0.0118** | 0.0138 | 0.0166 | **0.0132** |
| CRaWl | $0.0208 \pm 0.0022$ | $0.0197 \pm 0.0019$ | **$0.0095 \pm 0.0015$** | $0.0322 \pm 0.0013$ |

We stick to the experimental protocol suggested by Xu et al. (2019). Specifically, we perform a 10-fold cross validation. Each dataset is split into 10 stratified folds. We perform 10 training runs where each split is used as test data once, while the remaining 9 are used for training. We then select the epoch with the highest mean test accuracy across all 10 runs. We report this mean test accuracy as the final result. This is not the most realistic setup for simulating real world tasks, since there is no clean split between validation and test data. But in fact, it is the most commonly used experimental setup for these datasets and is mainly justified by the comparatively small number of graphs. Therefore, we adopt the same procedure for the sake of comparability to the previous literature. For COLLAB and IMDB-MULTI we use the same 10-fold split used by Zhang et al. (2018). For REDDIT-BIN we computed our own stratified splits. We also computed separate stratified 10-fold splits for hyperparameter tuning.

We adapt the training procedure of CRaWl towards this setup. Here, the learning rate decays with a factor of 0.5 in fixed intervals. These intervals are chosen to be 20 epochs on COLLAB and REDDIT-BINARY and as 50 epochs on IMDB-MULTI. We train for 200 epochs on COLLAB and REDDIT-BINARY and for 500 epochs on IMDB-MULTI. This ensures a consistent learning rate profile across all 10 runs for each dataset.

Table 9 reports the achieved accuracy of CRaWl and several key baselines on those datasets. For the baselines, we provide the results as reported in the literature. For comparability, we only report values for baselines with the same experimental protocol. On IMDB-MULTI, the smallest of the three datasets, CRaWl yields a slightly lower accuracy than most baselines. On COLLAB, our method performs similarly to standard MPGNN architectures such as GIN. CRaWl outperforms all baselines that report values for REDDIT-BIN. Note that GSN, the method with the best results on COLLAB and IMDB-MULTI, does not scale as well as CRaWl and is infeasible for REDDIT-BIN which contains graphs with several thousand nodes.

### B.4 Counting Substructures

We conduct additional experiments on the task of counting various subgraphs in synthetic graphs. To this end we utilize a benchmark dataset proposed by Chen et al. (2020). This dataset constitutes a graph-level regression problem that aims to infer how often each of the following subgraphs occurs in a given graph: Triangles, Tailed Triangles, 3-Stars, and 4-Cycles. The same dataset was used by Zhao et al. (2022) to evaluate GNN-AK, a state-of-the-art subgraph GNN. This experiment therefore offers a fine-grained comparison between both approaches.

We train CRaWl with 6 layers, hidden dimension 256 and window size 8 for 500 epochs and average the results over 4 models trained with different random seeds. Table 10 provides the test results. We report the MAE for each target subgraph. The baseline results were obtained from Zhao et al. (2022). Note that standard deviations were not reported.

As expected, both GNN-AK and CRaWl yield significantly better results on subgraph counting than simple MPGNNs like GCN and GIN. CRaWl yields the best results when counting stars, while GNN-AK achieves the lowest MAE on triangles and 4-cycles. Therefore, the structural information captured by CRaWl and subgraph GNNs like GNN-AK seems to capture different information have different strengths and weaknesses.

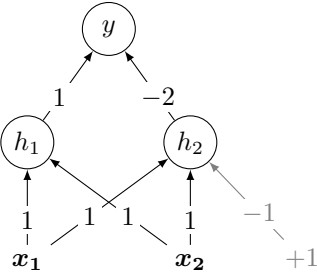

Figure 6: Implementing the XOR function using a neural network with ReLU activations

This overall result aligns with our main experiment in Table 1, where CRAWL outperforms subgraph GNNs on molecular data while GIN-AK+ yields better results on vision datasets.

## C  Additional Proof Details

### C.1  Detailed construction of the polynomial-size MLP

Finally, let us provide additional thoughts on Theorem 4. The provided proof is based on the assumption that our CNN filters can be an arbitrary MLPs. As MLPs are universal function approximators, they can theoretically memorize all feature matrices of a CFI graph $G_k$. This CNN filter enables CRAWL to distinguish $G_k$ from $H_k$, which $k$-WL fails to do. If we simply rely on the universality of MLPs in this simple argument, then the size and runtime of the required MLP grows exponentially in $k$. This observation has no effect on the correctness of the proof. It is also no limitation when comparing CRAWL to $k$-WL, since $k$-WL also has an exponential runtime for growing $k$ and only values of $k \leq 3$ have practical significance.

Nonetheless, with a more careful analysis it can be shown that the size of the MLP required in the proof has a polynomial upper bound. It is known that CFI graphs can be distinguished in polynomial time Cai et al. (1992). As for all computational problems in P, there do exist logical circuits of polynomial size that distinguish CFI graphs. In particular, for a CFI graph $G_k$ there exists a Boolean circuit that decides for a given adjacency matrix $A$ whether it corresponds to a graph that is isomorphic to $G_k$. It is also well known that Boolean circuits can be simulated with ReLU activated MLPs with polynomial overhead. For illustration purposes, we included the implementation of an XOR gate in Figure 6, the other Boolean operators can be implemented similarly. Therefore, we can simulate the Boolean circuit that detects $G_k$ with an MLP of a size that is polynomial in $k$.

