# OpenReview forum: "Walking Out of the Weisfeiler Leman Hierarchy: Graph Learning Beyond Message Passing"
_TMLR — Accepted by TMLR_

### Review · Reviewer_UkCv · 2023-05-16

**Summary Of Contributions:**

This paper proposes a new GNN, CRaWl, whole layers first sample random walkers from the input graph and then apply 1D-CNN to the encoded representations of subgraphs induced by the random walkers.
The expressive power of CRaWl is compared with that of the k-WL algorithm (and thus with Message Passing-based GNN). It is shown that these two are incomparable: On the one hand, there exist graph pairs that CRaWl can identify with window size and path length O(k^2) but not by k-WL. On the other hand, there exist graph pairs that can be easily identified by 1-WL but not by CRaWl with window size and path length O(n), where n is the number of nodes in a graph.
Finally, CRaWl is applied to synthesis and real graph classification problems to elucidate the prediction performance and robustness in terms of hyperparameters and random walk sampling strategies.

**Audience:**

Yes

**Broader Impact Concerns:**

There are no huge concerns about broader impacts.


**Claims And Evidence:**

Yes

**Requested Changes:**

I think the following questions are natural. From the perspective of novelty and significance, it would be desirable to answer these questions. However, as stated in TMLR's [acceptance criteria](https://jmlr.org/tmlr/acceptance-criteria.html), answering these questions is not mandatory.
- Does the representations obtained with CRaWl capture different features of graphs compared with MPNN representations?
- This paper does not compare the proposed method with the random walk-based GNNs introduced in Related Work. Is CRaWl's performance betternthan them?

Requested Changes
- P.2, Section 1.1: This paper refers to Wu et al. (2020) as a reference for Message Passing GNN. However, since MPNN was proposed by Gilmer et al. (2017), I think Gilmer et al. (2017) is appropriate.
- P.2, Section 1.1, Related Work mentioned existing studies on MPNNs, subgraph GNNs, and random walk-based GNNs. However, their relevance to this study needs to be discussed, at least explicitly. In particular, it would be better to discuss how the proposed methods are positioned in the line of research of random walk-based GNNs. For example, what problems can existing methods not solve, and how does the proposed method solve them?
- P.8, Theorem 2: It is claimed that $G$ and $G'$ cannot be "distinguished" by CRaWl of certain parameters. However, this concept is not mathematically defined. Does it means $\mathcal{X}(G) = \mathcal{X}(G')$? If so, I think it is clearer and more straightforward to state it directly.
- P.13, Table 1: What do boldface characters represent?
- P.13, Section 4.3: I suggest clarifying when and where the leaderboard was obtained.
- P.21, Appendix A.1: missing reference to a section.

**Strengths And Weaknesses:**

**Strengths**
- As opposed to other random-walk-based GNNs, the expressive power of the proposed method is provably beyond the WL hierarchy.
- Writing is clear. I can easily understand the main point of the paper.
- Reproducibility of numerical experiments is high.

**Weaknesses**
- Relation to other random-walk-based GNNs is not discussed in depth.
- The superiority of the proposed method in terms of implementation is not discussed (See Claim and Evidence)
- Performance gain of the proposed method empirically on node prediction tasks is moderate.

**Claim and Evidence**

This paper's main claims are as follows:
- CRaWl's expressive power for the graph isomorphism problems is incomparable to MPNNs.
- CRaWl's architecture suits today's computing environments, such as GPUs.
- CRaWl's empirical prediction performance is comparable to that of SOTA GNNs.

The first claim corresponds to Theorem 1. As far as I have checked the proof, this claim is correct.
The second claim is not numerically verified if I do not miss any information. However, since the architecture consists of random walk samplers and 1D-CNNs, I expect that the architecture is suitable for implementation on GPUs. Nevertheless, I am not sure this feature is a distinguished characteristic of the proposed method because we can implement usual MPNNs efficiently on GPUs for graph prediction problems where we have many small graphs.
The third claim has been tested on several types of graph datasets, including molecular datasets (MOLPCBA, ZINC), image datasets (MNIST, CIFAR10), and synthesis datasets (CSL). In addition, this paper compares with MPNN models mentioned in the related work. The code is provided as supplementary material explaining the dataset partitions and hyperparameters. Therefore, their results certainly justify the practical performance on graph prediction tasks. On the other hand, performance gain on node prediction tasks is limited (Appendix B.)

Considering the above points, I think the evidence generally supports these claims, although some parts are not verified.

**Audience**
In graph learning problems, GNNs are known to be a practical approach. Their theoretical and empirical expressive power is one of the central research topics in graph machine learning, especially GNNs beyond the WL algorithm framework. Since this paper proposes a GNN not inside the WL hierarchy, it interests the TMLR audience.

---

> ### Author Response · Authors · 2023-05-21
> **Response to Reviewer UkCv**
>
> We thank the reviewer for the positive and thoughtful feedback.
>
> Let us address the main questions and comments in the following.
>
> **Regarding hardware utilization**: \
> Random walks can be sampled either on the GPU or multi-threaded on the CPU while loading data.
> 1D-CNNs are of course highly optimized for GPUs. When implemented right, CRaWl will run with a GPU utilization close to 100% and have similar wall-clock runtime for training as advanced MPNN layers. We do not claim that this distinguishes CRaWl from MPNNs, which are of course also highly efficient on modern hardware. \
> We do provide some numbers comparing the two in Table 5. Note that the numbers provided for absolute train times of MPNNs are notoriously sparse in Graph Learning literature. The main point of the table is to demonstrate that training CRaWl does not take significantly longer than training MPNNs on similar hardware.
>
> **Regarding the relationship to Subgraph GNNs and Random Walk GNNs**: \
> CRaWl shares conceptual similarities to Subgraph GNNs, as the walklets can be viewed as subgraphs sampled from a distribution based on walks. However, the manner with which these subgraphs are embedded differs fundamentally, as CRaWl does not make use of MPNNs. Instead, for each subgraph (or walklet) we processes explicit binary encodings of the structure with an MLP that is implemented through a 1D-CNN. This methodological difference is the key reason for CRaWl's incomparability to the WL-hierarchy. \
> Random-walk-based GNNs like RAW-GNN aggregate node features along random walks using RNNs. Critically, these methods do not add explicit structural features and mostly aim at node- and link-level tasks where local structural detail is of lesser importance. In comparison, we focus on graph-level tasks where the detection of structural details is of high importance and a bottleneck of standard GNN architectures. Molecular data in particular constitutes a domain that CRaWl is very well suited for.\
> We also note that the additional experiments with CRaWl in Appendix B also use graph-level datasets, although these are small and lack standardized splits. We can highlight these differences more clearly in the revised version.
>
> **With regards to the features captured by the graph embeddings**:\
>  With the experiment on the CSL dataset we do demonstrate that CRaWl does learn to extract features that are not accessible to standard MPNNs. On real world datasets it is harder to quantify the degree to which the graph features captured in the embeddings overlap with those extracted by MPNNs. However, the strong predictive performance indicates that the features CRaWl extracts are at least as useful as those captured by MPNNs.
>
> **Regarding the question on Theorem 2:** \
> For this theorem $\mathcal{X}(G)=\mathcal{X}(G')$ holds and is the reason why CRaWl can not distinguish the graphs. However, there are cases where $\mathcal{X}(G)\neq\mathcal{X}(G')$ yet CRaWl will still fail to distinguish the graphs with high probability. We formalize this in the paragraphs after Theorem 2 and will try to highlight these definitions more clearly.
>
> **Additional Comments:** \
> In Table 1 we highlight the best result. Any other result with a mean inside the standard deviation is considered to be statistically identical and also highlighted.
>
> We will also incorporate the constructive suggestions on citations and references.

---

> > ### Comment · Reviewer_UkCv · 2023-06-04
> >
> > I thank the authors for answering my questions.
> >
> > **Regarding hardware utilization:**
> >
> > I do not think CRaWl should outperform usual MPNN regarding hardware utilization. It is sufficient that the proposed method is comparable to MPNN. I am on the same page with the authors in this regard. Also, I understand that Table 5 supports the authors' claim numerically (although the training time of CRaWl is longer than that of GIN since CRaWl's performance is better than GIN, we cannot directly compare the two.)
> >
> > **Regarding the relationship between Subgraph GNNs and Random Walk GNNs**
> >
> > I better understand how the proposed method differs from Subgraph GNNs and random-walk GNNs, respectively. Considering this information, I think it is worth numerically comparing the proposed and existing methods. At a high-level concept, by comparing with Subgraph GNNs, we can explore appropriate encoding methods for subgraph information. Similarly, by comparing with the random-walk-based method, we can check whether encoding structural information from random walks contributes to the predictive performance.
> >
> > **With regards to the features captured by the graph embeddings**
> >
> > I agree with the authors that the result of the CSL dataset supports the claim. If we want to strengthen the claim, another justification would be to check whether ensembling the proposed and MPNN-based method(s) enhances the predictive performance because it is known as ensembling (e.g., Kuncheva and Whitaker, 2003).
> >
> > [Kuncheva and Whitaker, 2003]: Kuncheva, L.I., Whitaker, C.J. Measures of Diversity in Classifier Ensembles and Their Relationship with the Ensemble Accuracy. Machine Learning 51, 181–207 (2003). https://doi.org/10.1023/A:1022859003006
> >
> > **Regarding the question on Theorem 2**
> >
> > OK. Thank you for adding the explanation.
> >
> > **Additional Comments**
> >
> > Thank you for your explanation. I would advise adding this explanation to the caption of Table 1.

---

> > > ### Author Response · Authors · 2023-06-06
> > > **Response to Reviewer UkCv**
> > >
> > > We thank the reviewer for the additional feedback. Let us add two further remarks.
> > >
> > > **Subgraph GNNs**
> > >
> > > We agree that further comparison to Subgraph GNNs is beneficial.
> > > We are adding an experimental evaluation on counting different subgraphs with CRaWl to the Appendix based on a suggestion by Reviewer P2Wb.
> > > There, we do compare CRaWl to GIN-AK, which is a state-of-the-art Subgraph GNN.
> > > The preliminary results are already available in the respective OpenReview discussion [here](https://openreview.net/forum?id=vgXnEyeWVY&noteId=cW27KcQXgd).
> > >
> > > The methods do indeed appear to have different strengths and weaknesses, as neither method is best across all considered subgraphs.
> > > This is also reflected by our main experiments in Table 1, where CRaWl has an edge on molecular data but GIN-AK yields slightly better results on CIFAR10.
> > > Such a direct comparison to random-walk-based methods is of lesser interest to us, as these do not primarily aim to capture the graph structure in fine detail.
> > >
> > >
> > > **Ensembling**
> > >
> > > Since our CRaWl layer is compatible with MPGNNs and also with Graph Transformers, it raises the question what would happen if we combine all three techniques as all three methods focus on different aspects of the graph structure.
> > > Through a combined architecture that for example runs all three aggregation methods in parallel we would thus expect a slight performance gain, unless the expanded model capacity results in overfitting.
> > >
> > > However, we believe that doing such experiments thoroughly will be a project in its own right.
> > > For the CSL dataset, one clearly does not need more than CRaWl as this method already performs perfect predictions.
> > > Evaluating ensembles here would mean that a model simply learns to ignore all MPGNN predictions and rely exclusively on CRaWl embeddings.
> > > Only for other datasets such as substructure counting, ensembling could be used to study whether the different views of CRaWl, MPGNNs, and Graph Transformers are synergetic.
> > >
> > > We will highlight this aspect more prominently in the outlook section of the revised paper.

---

> > > > ### Comment · Reviewer_UkCv · 2023-06-08
> > > >
> > > > I thank the authors for the response.
> > > >
> > > > **Subgraph and Random-walk-based GNNs**
> > > >
> > > > I thank the authors for adding numerical evaluations for GIN-AK. I think it would be a nice direction of research to elucidate where the performance differences between CRaWl and GIN-AK come from, although both methods aim to extract the subgraph information. Regarding the acceptance criteria, I think it does not significantly incur the paper's claim at least comparable to the SOTA (ref. Claim and Evidence section of my initial comment.)
> > > > Regarding the comparison with random-walk-based methods, in my opinion, we may obtain insights from the comparison with them: for example, whether the proposed method can effectively extract the structural information of subgraphs induced by random walks. Nevertheless, as I wrote in my initial comment, I think this is not a must, given TMLR's acceptance criteria.
> > > >
> > > > **Ensembling**
> > > >
> > > > I agree with the authors that it depends on tasks whether both representations obtained by CRaWl and MPNN are required. It would be a future research direction in what condition we can enjoy the benefit of ensembling.

---

### Review · Reviewer_3MCo · 2023-05-17

**Summary Of Contributions:**

This paper proposes CRAWL, a model for generating graph representations. The model simulates random walks and then applies a CNN to the subgraphs induced by nodes that appear along each walk. The authors study the expressive power of CRAWL and they show that the proposed model is not comparable to standard message passing GNNs in terms of expressiveness. There exist graphs that can be distinguished by CRAWL but not by standard GNNs and there also exist graphs that can be distinguished by standard GNNs but not by CRAWL. The model is evaluated in graph classification and graph regression tasks where it outperforms the baselines in several cases.

**Audience:**

Yes

**Broader Impact Concerns:**

-

**Claims And Evidence:**

Yes

**Requested Changes:**

The following adjustments are related to the weaknesses listed above:

- Make clear that the proposed model processes induced subgraphs and provide more details about that. Also, discuss how the proposed model is related to subgraph GNNs.

- Explain how one can choose the values for the different hyperparameters. Is there some relatively cheap heuristic one could employ to select these values?

- Since the model is supposed to capture long range interactions between nodes, experiment with such datasets (e.g., Long range graph benchmark).

- Improve the proof of Theorem 1. For instance, explain why the CNN (which in this case is equivalent to a fully-connected layer in my understanding) can approximate the function that checks whether a feature matrix belongs to $R_G$.

Minor:

- Typos:\
p.6: "graph-evel" => "graph-level"\
p.7: "edge set" => "node set"\
p.7: "node set" => "edge set"\
p.10: "length s=" => "length l="\
p.10: "graphs exists" => "graphs exist"

- In many cases throughout the paper \citeyear is used instead of \cite (for instance, in second paragraph of Related Work subsection). This should be fixed in the revision.

- I would suggest the proofs of Theorem 1 are moved to the appendix to help improve the paper's readability

**Strengths And Weaknesses:**

Strengths:

- The proposed model is conceptually very simple, but also novel to the best of my knowledge. In my view, it is a useful addition to the literature of GNNs since it is different from most existing models that are instances of the message passing architecture, while random walks have not been fully explored in this context.

- The empirical results are interesting. The proposed model yields good performance on most datasets. Also the ablation study is interesting since it seems that on most datasets, the adjacency and identity encodings improve significantly the performance of the proposed model.

- The paper is well-written and easy to read, while it also has some nice figures to illustrate the proposed model.

Weaknesses:

- Even though the proposed approach simulates random walks over the graph, it appears that walks themselves cannot capture structural properties of the graph, while they also empirically seem to yield inferior performance when used on their own. This is why the authors introduce the adjacency and identity encodings. But then, one could think of the feature matrices as connected subgraphs sampled from the graph. I would like the authors to comment on that and also discuss how the proposed model is related to subgraph GNNs.

- To apply the proposed model to some dataset, one needs to specify the values of several hyperparameters (e.g., length of walk, window size, etc.). Is there a principle or rule of thumb for selecting the values such that the desired performance is achieved? Did the experiments on the real-world datasets provide any insights? This would be much appreciated by a practitioner who does not have the time to experiment with different combinations of hyperparameter values and does not also have any knowledge about how to choose them.

- The authors claim that the proposed model detects long range interactions between nodes. However, this is shown neither theoretically nor experimentally. There are now available some datasets that require a method to perform long range interaction reasoning in order to achieve strong performance [1]. I would suggest the authors evaluate the proposed model on some of these datasets.

- The proposed model achieves competitive results on ZINC and MOLPCBA, but on the 3 considered TUDatasets (results reported in Table 6) its performance is not that impressive. Is there a specific reason behind that? The number of samples of these datasets is much smaller than the number of samples of ZINC and MOLPCBA. Could this be the reason for the the performance drop?

- The proof of the first part of Theorem 1 makes some assumptions that do not make fully sense and in my view, the authors should provide some explanations.
	- First of all, the considered instance of the model is very different from the model one would use in a real-world application. The authors assume that both the walk length and the window size are equal to $12n^2$. That would mean that the number of elements of the feature matrix is in the order of $\mathcal{O}(n^4)$ which would be prohibitive for real graphs.
	- Then, the authors claim that by the universality of the CNN filters, the model can approximate function $f$. However, if I am not wrong, in this case, the CNN is equivalent to a fully-connected layer that maps the feature matrix into a scalar. Two walks that visit the same nodes in the opposite order encode the same information but the emerging feature matrices are very different from each other. This is not captured by a typical CNN which does not exhibit such invariance properties. Moreover, the cardinalities of sets $R_G$, $R_H$ and $S$ are very large (since different walks give rise to different feature matrices) and the components of their elements are not continuous. Thus, I really doubt a fully-connected layer can learn to approximate function $f$.

[1] Dwivedi, V. P., Rampášek, L., Galkin, M., Parviz, A., Wolf, G., Luu, A. T., & Beaini, D. (2022). Long range graph benchmark. Advances in Neural Information Processing Systems, pp. 22326-22340.

---

> ### Author Response · Authors · 2023-05-21
> **Response to Reviewer 3MCo**
>
>
> We thank the reviewer for the positive and constructive feedback.
>
> Let us address the main comments and questions in the following.
>
> **Regarding the relationship to subgraph GNNs:** \
> The sampled walklets can be viewed as subgraphs. Therefore, there are methodological similarities to subgraph GNNs.
> The key difference are the features we construct for each walklet (or subgraph) and the way these features are embedded.
> Subgraph GNNs process subgraphs with MPGNNs, thereby tying their expressiveness to the WL-hierarchy. \
> We instead define complete binary encodings of the subgraph structure which are effectively embedded by an MLP implemented through 1D-Convolutions.
> Our representation is non-unique and depends on the vertex order with which the subgraph is randomly traversed.
> This major methodological difference is what separates CRaWl from the entire WL-hierarchy.\
> We will emphasize the relation to subgraph GNNs more clearly in the final version.
>
> **Regarding Hyperparameter Tuning:** \
> While some tuning is always needed for optimal performance, we found our default configuration of walk length 50 and window size 8 to be a robust initial setting across all datasets.
> A window size of at least 6 is critical on molecular data to capture organic rings, as shown on Section 5.1.
> If overfitting is observed, then reducing the number or length of walks is a simple and effective measure to improve performance.
> During validation we observed monotonic improvements for increasing walk lengths, with diminishing returns above $\ell=150$.
> Therefore, a simple heuristic for evaluation is to increase the walk length to the maximum that fits into GPU memory. \
> We will highlight these heuristics in the revision.
>
> **Regarding the Proof of Theorem 1:** \
> The parameter setting is an artifact of the proof. The $k$ is the WL-dimension and hence usually not large, the relevant values for $k$ are 2,3 (since k-WL with large k is prohibitive anyway). The construction already works on graphs of size $\mathcal{O}(k^2)$, but of course the theorem then also also holds for graphs of size n much larger than $k^2$. \
> Having said that, clearly this is a theoretical result, for larger values of $k$ we will not be able to learn such filters. However, for larger $k$, k-GNNs are also a purely theoretical idea because they need $n^k$ nodes.
>
> **Additional Comments:** \
> We will evaluate CRaWl on LRGB and add the results to the revised version. On the TUDatasets the size of the datasets is likely the main reason for the results. As CRaWl is able to detect local structures in fine detail it is also able to overfit when few training graphs are present.
>
> We will also address the typos and citation issues that were kindly pointed out.

---

> ### Author Response · Authors · 2023-05-30
> **Results on LRGB Datasets**
>
> Dear Reviewer,
>
> we have conducted experiments on 3 datasets from the Long-Range Graph Benchmark and obtained the following preliminary results:
>
> | Model                   | PASCAL-VOC (F1) | PEPTIDES-FUNC (AP)  | PEPTIDES-STRUCT (MAE) |
> |-------------------------|-----------------|---------------------|-----------------------|
> | GCN                     | 0.1268 ± 0.0060 | 0.5930 ± 0.0023     | 0.3496 ± 0.0013       |
> | GatedGCN         | 0.2873 ± 0.0219 | 0.6069 ± 0.0035     | 0.3357 ± 0.0006       |
> | Transformer             | 0.2694 ± 0.0098 | 0.6326 ± 0.0126     | 0.2529 ± 0.0016       |
> | SAN                     | 0.3230 ± 0.0039 | 0.6439 ± 0.0075     | 0.2545 ± 0.0012       |
> | CRaWl                   | **0.4588 ± 0.0079** | **0.7074 ± 0.0032** | **0.2505 ± 0.0029**       |
>
> The baseline results are obtained from LRGB paper. We believe these results validate our claim that CRaWl captures long-range interactions and will add this experiment to the paper. We thank the reviewer for the very constructive suggestion.

---

### Review · Reviewer_P2Wb · 2023-05-28

**Summary Of Contributions:**

The authors propose a framework to process graph data based on random walks. The advantage of the method is both theoretical and practical. The authors theoretically show that the proposed mechanism goes beyond the WL test and its expressiveness. Also, using the proposed method allows to utilize existing 1D convolution code that is highly optimized by different libraries such as cuda.

The authors also show the strength of their method on several graph classification and regression tasks, showing significant improvement over baseline methods.



**Audience:**

Yes

**Broader Impact Concerns:**

I do not see any potential negative outcomes of this work.

**Claims And Evidence:**

Yes

**Requested Changes:**

I would like to see the authors respond to points 1-3 in my 'weaknesses' report. As stated in my review, I think that after appropriately addressing these points, the paper should be accepted.

**Strengths And Weaknesses:**

Strengths:

1. The paper is well motivated and easy to follow.

2. The paper extends on ideas that allow to utilize efficient 1D operations in cuda, such as [1],[2]. The extension is both theoretical and technical, so the contribution is significant.

3. The paper experiments with several datasets and shows a remarkable improvement over baselines methods.

4. The authors also conduct an ablation study to learn about the influence of the hyperparameters, such as the length of the walk.

Overall, I think that the paper is very solid and should be accepted, after minor changes.

Weaknesses:

1. The authors lack some background material such as a comparison with earlier models that propose a similar approach (although different, and the proposed method significantly extends them), such as [1],[2].

2. There are several typos and broken references throughout the paper - additional passes on the text should clear them out.

3. With respect to the theoretical representation power of the proposed method, I think it would be beneficial if the authors experiment and report the obtained MAE on the substructure counting dataset from GIN-AK (which is already cited and compared with by the authors)

[1] Path Integral Based Convolution and Pooling for Graph Neural Networks

[2] pathGCN: Learning General Graph Spatial Operators from Paths

---

> ### Author Response · Authors · 2023-05-31
> **Response to Reviewer P2Wb**
>
> We thank the Reviewer for the constructive suggestions.
>
> We will add the missing comparison and fix the mentioned typos and references.
>
> We are also currently evaluating CRaWl on the substructure counting dataset from GIN-AK. We will report the results as soon as they are available.

---

> ### Author Response · Authors · 2023-06-05
> **Results on Substructure Counting**
>
> Dear Reviewer,
>
> as suggested, we have conducted an experimental evaluation on the substructure counting dataset from GNN-AK.
>
> we report the MAE for each target. The baseline results are obtained from the GNN-AK paper (Zhao et al. 2022):
>
> | Model    |       Triangle        | Tailed Tri.     | Star                | 4-Cycle         |
> |:----------|:-----------------:|:-----------------:|:---------------------:|:-----------------:|
> | GCN      | 0.4186          | 0.3248          | 0.1798              | 0.2822          |
> | GIN      | 0.3569          | 0.2373          | 0.0224              | 0.2185          |
> | PNA      | 0.3532          | 0.2648          | 0.1278              | 0.2430          |
> | GCN-AK+  | 0.0137          | 0.0134          | 0.0174              | 0.0174          |
> | GIN-AK+  | 0.0137          | **0.0112**      | 0.0150              | 0.0150          |
> | PNA-AK+  | **0.0118**      | 0.0138          | 0.0166              | **0.0132**      |
> | CRaWl    | 0.0208  | 0.0197  | **0.0095** | 0.0322 |
>
>
> Both GNN-AK and CRaWl outperform MPNNs by a significant margin, as expected.
> However, which method works best depends on the target.
> GNN-AK seems to be slightly better at counting cycles and its performance depends on the MPNN architecture used as a kernel.
> In contrast, CRaWl has an edge when counting stars.
>
> We note that this experiment lacks a standardized setting and the literature does not report the standard deviation.
> Nonetheless, these results do provide a nuanced view on each methods ability to count certain subgraphs and are a valuable addition.
> We will add these results to the appendix and add a reference in the main paper.
>
> We would like to thank the reviewer for this very constructive suggestion.

---

### Review · Reviewer_9Kh9 · 2023-05-28

**Summary Of Contributions:**

This work proposes a new neural network-based approach to perform graph representation learning. The approach first samples random walks from the graph, and then constructs a feature matrix for each sampled random walk. The work proposes to use CNNs to encode these sampled walks. The authors prove that this method can distinguish connected graphs when the sampled walks are long enough ($O(|V||E|)$) and are sufficiently many ($O(|V|)$). Therefore, it can distinguish graphs that k-WL cannot distinguish. The proposed method also fails to distinguish graphs when the samples walks are short and only a few. Also, the proposed method also fails when the graphs are not connected. Experiments show the proposed method can outperform previous methods in some settings.

**Audience:**

Yes

**Claims And Evidence:**

Yes

**Requested Changes:**

1. Address weakness 1 by characterizing the number of walks is needed to surely or with high probability to distinguish non-isomorphic graphs. Make sure the statement is rigorous.

2. Address weakness 3 by Giving sufficient credits to previous works that sample and encode random walks instead of standard graph neural networks for graph representation learning.

**Strengths And Weaknesses:**

Strengths:
1. The work is well written and motivated well.
2. The explanation on the theory is clear and extensive. I appreciate that the authors transparently show strengths and limitations of the proposed method.
3. Experiments show the advantages of the proposed approach, e.g., over the datasets MOLPCBA and CSL.

Weaknesses:
1. Although the explanation on the theory is clear, I have some concerns on the value of the theory. By checking the proof of theorem 3, if I understand correctly, the proof is actually not only suitable for the graphs that k-WL fails to distinguish. It can also be applied to any connected non-isomorphic graph pairs, as long as the walks have length O(n^3) (suppose n many nodes and n^2 many edges). The only condition needed here is that graphs are connected. This poses a concern that how this algorithm can be so powerful. With some further thoughts, I noticed that there are some missing logical components in the proof.

a) First, the authors did not prove why the needed sample size m is O(n). The proof asks for sufficiently many sampled walks ("last sentence in Sec. 3.1").

b) Second, the proof logic is to leverage the fact (observation 1) that if the structure feature matrix X keeps the same between two graphs, the two graph will be isomorphic. This matches expectation. Essentially, the random walk associated the nodes with an order which determines the row indices in X. Then, the comparison between two unlabeled graphs reduces to comparing two labeled graphs. Of course, if the latter two are isomorphic, the former two are isomorphic. But to surely demonstrate the former two are non-isomorphic, one should enumerate all possible orders of the nodes, i.e., the walk samples in the order O(n!). I am not sure how the authors can prove that only O(n) many walks are sufficient. Moreover, even if O(n) many walks can determine it with high probability, theorem 3 should be claimed rigorously, i.e., with a language in probability instead of surely.

2. The proposed method is kind of complex. As observed above in the theory, essentially, the algorithm uses sampled walks to determine an order of the nodes. Each constructed matrix will be at least of n*n sizes. So now, each walk becomes a n*n matrix that needs a CNN to encode. Also, as suggested by the paper, the number of walks should be at least O(n). So, the overall complexity is much higher than MPNN.

3. Some missing references: Using sampling walks and encoding walks for graph representation learning are not new. The local identity idea read similar to those adopted by anonymous walks [1], which have been adopted to represent graphs based on neural network encoding [2,3]. Using CNNs on random walks to represent graphs has also been considered before [4].

[1] Anonymous Walk Embeddings
[2] Inductive Representation Learning in Temporal Networks via Causal Anonymous Walks
[3] Algorithm and System Co-design for Efficient Subgraph-based Graph Representation Learning
[4] Path Integral Based Convolution and Pooling for Graph Neural Networks

---

> ### Author Response · Authors · 2023-06-01
> **Response to Reviewer 9Kh9**
>
>
> We thank the Reviewer for the feedback and questions.
>
> We will of course update the related work section to include the mentioned works and make clear how our approach differs from previous work. Thanks a lot for pointing those out.
> When skimming over reference [4] (*Path Integral Based Convolution and Pooling for Graph Neural Networks*), this looks more like an interesting variant of a multi-hop MPGNN than something actually exploiting random walks or CNNs. Please correct us, if we are wrong here.
>
> Regarding the theory questions:
>
> **1.**
> You are right, with sufficiently large window size, in principle CRaWl will be able to distinguish any two graphs.
> However, the size of the MLPs in the CNN filters will have to be very large (exponential in the graph size), so this does not scale.
> However, for the comparison with k-WL note that the relevant values of k are small, because k-WL quickly becomes completely intractable for k larger than, say, 3.
> In Theorem 3, we can choose both the window size and the walk length in $\mathcal{O}(k^2)$ and still distingush arbitrarily large graphs that k-WL does not distinguish.
>
> **1a.**
> In the proof we employ a standard Monte-Carlo sampling argument.
> We have shown that CRaWl distinguishes the graphs based on a single random walk with probability at least $\frac{1}{2}$.
> In order to distinguish the two graphs with probability at least $1-\epsilon$, we need to use at least $m \geq \log \frac{1}{\epsilon}$ repetitions or random walks.
> The theorem is indeed better phrased in terms of probabilities and we will update it accordingly.
>
> **1b.**
> You are right that there is a hidden exponential dependence on k, but this is in the universality assumption for the CNN filters and not in the number of samples.
> To distinguish nonisonorphic graphs G and H, the function the filters need to learn is "there is a copy of G in my window (in any order)".
> Arguably, the universality assumption is not practical, but it is often made in expressiveness results (for example, Xu et al. (2019) on GNNs and WL).
>
> Actually, in our specific setting, we could avoid the universality assumption and the exponential blow-up.
> It is known that the graphs coming out of the CFI construction can be distinguished by a polynomial size circuit, and this circuit can be emulated by the MLP in the CNN filter.
> However, carrying this out in detail is quite complicated (and would require a refined model of the CNN filters).
> It does not seem this is worth the effort, because in the end the result remains the same, and the intuitive argument is also the same, so this would not give any new insights.
>
> **2.**
> The walk feature matrices are of size $\mathcal{O}(\ell s)$ and in practice the window size $s$ is typically much smaller than $n$ with typical values below 10.
> The asymptotic runtime is higher than that of MPNNs, but this is a common drawback of many architectures stronger than 1-WL.
> We are expanding the section "Asymptotic Runtime" in the revised version to clarify this further.

---

> > ### Comment · Reviewer_9Kh9 · 2023-06-11
> > **Thanks for the detailed response**
> >
> > Thanks for the authors' response. It is great to see your designed walks of length $O(k^2)$ may avoid the complexity of k-subgraph isomorphism used in k-ML. Also, thanks for the update of the draft to make the statement of Theorem 3 more rigorous.
> >
> > I still have one remaining concern: I agree that by using m sampled walks, the proposed method can distinguish two graphs with high probability $(1-2^{-m})$. Although conceptually, using the walks paired with the structural encoding indeed give stronger expressive power than k-WL, does your result stated in Theorem 3 in stochastic sense really means stronger expressive power of your method?
> >
> > Think about the following experiments: There are two non-isomorphic graphs $G_1$ and $G_2$ which k-WL cannot distinguish. Then, one uses the proposed approach over these two graphs. One randomly picks one single node from either graph and randomly sample one walk with some length (say $k^2$) starting from either node. I believe the two sampled walks will be different with high probability due to the randomness of the sampling procedure. But is it really useful? The detected difference comes from randomness in sampling instead of detecting actual difference between the two graphs. So, I still have some concern on the statement. I think a better statement is needed to show the actual more expressive power.

---

> > > ### Author Response · Authors · 2023-06-12
> > > **Response to Reviewer 9Kh9**
> > >
> > > We thank the reviewer for the additional feedback. Regarding the question on the proof of Theorem 3:
> > >
> > > It does not matter that the sampled walks are different with high probability as long as both walks reach every single node of the graph at least once. The function $f$ the convolution filter learns is defined by $f(x)=1$ if $x$ contains the encoding of a copy of $G_1$​ in **any** order, and $f(x)=0$ otherwise.
> > > Here it is crucial that the function is really independent of the order of the nodes - whenever all of $G_1$​ is encoded, then we have $f(x)=1$ and otherwise $f(x)=0$.
> > >
> > > When $x$ is the sliding window of a walk on graph $G_1$​​, then with positive probability (namely when the random walk reaches every node of the graph at least once), $f(x)=1$. On the graph $G_2$​​, we always have $f(x)=0$ because it is non-isomorphic to $G_1$​ and all edges and non-edges of those graphs are encoded at least once in the encoding. Clearly, the whole statement depends on $f$ being 1 if and only if the input encodes $G_1$​ in any order.

---

> > > > ### Comment · Reviewer_9Kh9 · 2023-06-13
> > > > **Thanks for the detailed response**
> > > >
> > > > Thanks for the response. I think the problem is that the argument relies on the function f, which cannot be efficiently evaluated in practice. The function f should be aware of all possible walks' encoding matrices X sampled from either graph.
> > > >
> > > > I think the condition on the awareness of this function should be stated clearly.

---

> > > > > ### Author Response · Authors · 2023-06-15
> > > > > **Response to Reviewer 9Kh9**
> > > > >
> > > > > In our setting the function $f$ is implemented as an MLP. Therefore, evaluating $f$ would be require as much time as a forward pass of this MLP, which is polynomial in the number of parameters.
> > > > >
> > > > > Of course the number of parameters necessary to fit the function $f$ with an MLP grows quickly in $k$. However, as stated before, the practically relevant values are $k≤3$ as $k$-WL also scales poorly for larger $k$.
> > > > >
> > > > > We will add further clarification on these points to the paper.

---

> > > > > > ### Comment · Reviewer_9Kh9 · 2023-06-16
> > > > > > **Thanks for the detailed response**
> > > > > >
> > > > > > Thanks for the response. I did not talk about how to implement $f$ in a practical task. To make the theorem correct, such $f$ should be aware which graph each walk with its encoding matrix X is sampled from, which is with exponential complexity. Otherwise, the proof does not work.

---

> > > > > > > ### Author Response · Authors · 2023-06-16
> > > > > > > **Response to Reviewer 9Kh9**
> > > > > > >
> > > > > > > We thank the reviewer for the additional remarks.
> > > > > > >
> > > > > > > f is just a function from $ \lbrace 0,1\rbrace^r $ to $ \lbrace 0,1 \rbrace $, where $r=\mathcal{O}(s^2)$ is the length of the encoding of the graph in the window of the convolution filter. Since the domain $ \lbrace 0,1\rbrace^r $ is a compact set, this function can be approximated arbitrarily precisely by a multi-layer perceptron, which we use as convolution filter. The size of the MLP has no bearing on the correctness of the theorem. (We explained in an earlier answer [here](https://openreview.net/forum?id=vgXnEyeWVY&noteId=nXXC0v6Xjl) that by just applying the universal approximation theorem, the size of the MLP may be exponential in the input dimension $r$, but that this can be avoided for the specific graphs that we use here.)
> > > > > > >
> > > > > > > That's all we need. We're not sure what it means that "$f$ should be aware which graph each walk with its encoding matrix X is sampled from".

---

> > > > > > > > ### Comment · Reviewer_9Kh9 · 2023-06-16
> > > > > > > > **Thanks for the response**
> > > > > > > >
> > > > > > > > Universal approximation theorem does not implies this $f$ can be implemented with polynomial operations. The point is that the f should be $f=1$ if $X$ is sampled from a graph G. This assumption means that $f$ is aware of all possible X's sampled from a graph G. The computation of a function that satisfies this property may be exponential in the number of nodes (walk lengths). Of course, it is unclear whether using MLP to implement such $f$ has just a polynomial number of neurons. However, k-WL tests with constant k only need polynomial number of operations (poly in the number of nodes).
> > > > > > > >
> > > > > > > > Give you another example: each un-attributed graph with $n$ size can be represented via a discrete embedding in $\{0,1\}^{n^2}$. Graph isomorphism test is a function $g$ from $\{0,1\}^{n^2} \rightarrow $ isomorphic-equivalent classes. So, can $g$ be implemented via NN with a poly number of operations? Of course no (at least to the best of our current knowledge). Can MLP universally approximate it? Of course it can. So just saying universal approximation has no guarantee on the complexity.

---

> > > > > > > > > ### Author Response · Authors · 2023-06-19
> > > > > > > > > **Response to Reviewer 9Kh9**
> > > > > > > > >
> > > > > > > > > For any fixed $k$ we need to memorize all feature matrices of one CFI graph $G_k$.
> > > > > > > > > As mentioned before, the size of the MLP needed for this can be shown to be polynomial (see below for more details).
> > > > > > > > > Note that some specific graphs, such as CFI graphs, do allow for a polynomial sized MLP to memorize all representations.
> > > > > > > > > This may not be possible for arbitrary graphs, but CFI graphs are not arbitrary.
> > > > > > > > >
> > > > > > > > > However, as we argued before, the number of parameters needed for $f$ has no effect on the correctness of the proof, which is why we omit this complicated detail in the paper.
> > > > > > > > > Furthermore, if we consider a constant $k$, then the graph $G_k$ is fixed and the MLP needed to memorize all representations of $G_k$ is also constant in size and runtime, regardless of how the size depends on $k$.
> > > > > > > > > The walk length $\ell=\mathcal{O}(k^2)$ is also constant in this case and processing each walk would have constant runtime. Therefore, we would have an overall runtime of CRaWl that is linear in the number of walks $m$ (assuming fixed $k$).
> > > > > > > > > When fixing $k$ in this setting, both $k$-WL and CRaWl have polynomial runtime. However, CRaWl can distinguish $G_k$ and $H_k$ with high probability while $k$-WL will always fail to do so.
> > > > > > > > > We are therefore unsure what exactly the Reviewer means by "k-WL tests with constant k only need polynomial number of operations".
> > > > > > > > >
> > > > > > > > > #### *On the Polynomial MLP Size:*
> > > > > > > > > The trick here is that for the theorem, we only need to distinguish graphs coming from the CFI construction which allows us to avoid the exponential growth.
> > > > > > > > > The paper [1] details how to construct a polynomial-size circuit consisting only of XOR gates that can distinguish CFI graphs.
> > > > > > > > > This result easily carries over to our setting as an MLP with ReLU activations is able to compute the XOR function.
> > > > > > > > > This way we would have a polynomial-sized MLP (using more than one hidden layer) that can act as our filter and simulate this circuit.
> > > > > > > > >
> > > > > > > > > [1] Choiceless Polynomial Time, Symmetric Circuits and Cai-Fürer-Immerman Graphs https://arxiv.org/pdf/2107.03778v1.pdf

---

> > > > > > > > > > ### Comment · Reviewer_9Kh9 · 2023-06-19
> > > > > > > > > > **Thanks for the response**
> > > > > > > > > >
> > > > > > > > > > As we discussed, to distinguish graphs (k-WL fails to do), the sampled walks in the work have to be of length $l=\Omega(n^2)$, i.e., at least the squared number of nodes in the graphs. I suspect the function $f$ needs exponential operations w.r.t. $l$, i.e., exponential in $n^2$, which makes the theorem not that useful.
> > > > > > > > > >
> > > > > > > > > > If the proof idea in the reference can be used to demonstrate that such an $f$ can be implemented with polynomial operation, that will be great. I appreciate this observation. To make the work complete, please provide a proof in the manuscript that $f$ can be evaluated in poly time w.r.t. the number of nodes in the graph.

---

> > > > > > > > > > > ### Author Response · Authors · 2023-06-21
> > > > > > > > > > > **Response to Reviewer 9Kh9**
> > > > > > > > > > >
> > > > > > > > > > > We will add a detailed explanation of the polynomial construction to the manuscript.
> > > > > > > > > > > As the correctness of Theorem 3 does not strictly depend on this detail we will add it to the appendix and add a reference in the main section.
> > > > > > > > > > > Below, we provide a preliminary version of this detailed explanation.
> > > > > > > > > > >
> > > > > > > > > > > #### **Detailed construction of the polynomial-size MLP**
> > > > > > > > > > > Let us formalize how the XOR circuit constructed by [1] can be computed by an MLP with only polynomial size.
> > > > > > > > > > >
> > > > > > > > > > > Implementing XOR in feedforward neural networks with ReLU ist not too hard (see illustration [here](https://i.postimg.cc/6Ts9NcMQ/gadget.png)). If we can ignore all zero-weight operations, we trivially have a polynomial-sized feedforward neural network for the actual computation. When counting those empty operations (i.e. embedding the circuit into a standard feedforward neural network) we observe that the maximal width of each layer is polynomial. Thus the maximal number of edges between every two layers is still polynomial (squaring the width). Again multiplying with the depth (which is polynomial as the number of operations is polynomial) gives us a (loose but polynomial) upper bound on the size of such a network.
> > > > > > > > > > >
> > > > > > > > > > > In addition to the actual computation, one needs to convert the input to _any_ adjacency matrix of the graph. This is a simple operation as it effectively just means copying the bits of our representation to their correct positions. By using the nodes in the order of their first occurrence in the random walk, one can come up with a "gate" that selects for the i-th row of the adjacency matrix the corresponding row in our representation. Since that will fill the lower triangle of the adjacency matrix. By copying that information to the upper triangle one can complete the matrix. The key operation is thus to count until n ignoring "duplicate rows" which are explicitly indicated by a bit in the identity features of our representation. Counting up to a fixed n is possible using a polynomial number of logical operations (each of them can be encoded in a similar way to XOR from above).
> > > > > > > > > > >
> > > > > > > > > > > Executing those two polynomial-sized MLPs sequentially results again in polynomial a polynomial MLP. This finishes the proof sketch. To make it a complete formal proof, one only needs to write down the circuits that convert our representation into an adjacency matrix.
> > > > > > > > > > > While this is clearly possible (its just a combination of counting zeros in specific locations and then copying the corresponding bit if the count comes out right), writing down those formulas/circuits explicitly will not further understanding of the proof. Essentially every bit of the adjacency matrix will have some polynomial circuit attached to it for every possible bit that it may become. This is clearly still polynomial.
> > > > > > > > > > >
> > > > > > > > > > > [1] Choiceless Polynomial Time, Symmetric Circuits and Cai-Fürer-Immerman Graphs https://arxiv.org/pdf/2107.03778v1.pdf

---

> > > > > > > > > > > > ### Comment · Reviewer_9Kh9 · 2023-06-24
> > > > > > > > > > > > **Thanks for the response**
> > > > > > > > > > > >
> > > > > > > > > > > > Thanks for the further explanation. Although I do not fully get the proof, I am fine to accept the authors' argument. It is a little bit surprising to see there exist a polynomial-sized MLP that can recognize random walks from any graphs accurately. It would be great if the authors can incorporate such a discussion in the paper.

---

### Review · Reviewer_RF6X · 2023-05-29

**Summary Of Contributions:**

This paper introduces CRaWl, a method that convolves over random walks sampled from an input graph, in order to perform graph representation learning tasks. Importantly, the authors are able to show and prove that their layer is a fundamentally new approach with respect to expressivity ladders: the set of graphs it can and cannot distinguish cannot be reasoned about using the popular Weisfeiler-Leman hierarchy.

**Audience:**

Yes

**Broader Impact Concerns:**

No concerns.

**Claims And Evidence:**

Yes

**Requested Changes:**

See the weaknesses section above.

To further specify the first bullet point, if possible, please provide running times for more baseline models, on more datasets, to expand the findings of Table 5. If such numbers are not publicly available, the authors could consider making their own implementation.

**Strengths And Weaknesses:**

On the positive side:
* I believe it is very important to explore expressivity hierarchies other than WL. I applaud the authors for proposing a simple and elegant method that does exactly this.
* The proposed method, CRaWl, is certainly capable of extracting additional meaningful structure from the graph, and therefore clearly ticks the "1-WL outperformance" box that these papers should all have.
* The theoretical work is solid and well-motivated.

On the negative side:
* The scalability analysis of the model could have been executed better. Specifically, Table 5 is quite uninformative as it stands, with non-CRaWl numbers reported very scarcely. Given the model's very different mode of operation compared to a traditional GNN, It would be very useful to supplement the computational complexity analysis with additional runtime comparisons.
* When discussing the "asymptotic runtime", the authors write _"Let us emphasize that this [superlinear complexity] is a common drawback of architectures which are not bounded by 1-WL"_. In my opinion, this claim needs to be better qualified to avoid confusion. For example, even computing very rudimentary structural features, or even purely random node features (Sato et al.), already leads to being able to distinguish more graphs than 1-WL. The complexity of such approaches is linear in the graph size. I invite the authors to better qualify this statement.

---

> ### Author Response · Authors · 2023-05-31
> **Response to Reviewer RF6X**
>
> We thank the Reviewer for the positive and helpful response.
>
> We agree that Table 5 is to sparse and we will extend it with additional baseline results.
>
> We will also expand the section on the asymptotic runtime as suggested.
> While random node features (Sato et al.) boost theoretical expressiveness with minimal overhead, the performance gain observed on real-world classification tasks is marginal.
> A good example of precomputed structural features with significant performance gains on real-world datasets is GSN (Bouritsas et al.).
> However, its (worst case) precomputation runtime is in $\mathcal{O}(|V|^k)$, where $k$ is the size of the considered subgraphs.
> Higher order $k$-GNNs (Morris et al.) have a runtime in $\mathcal{O}(|V|^k)$ and Graph Transformers (Dwivedi et al.) usually require $\mathcal{O}(|V|^2)$.
> We can add a more precise discussion on this to the revised paper.

---

### Author Response · Authors · 2023-06-07
**Revision of the Paper**

Dear Reviewers,

we have just updated the paper with a revised version based on the helpful feedback.

The key changes are as follows:
1. We expanded the related work section and discuss the relationship to random walk GNNs and subraph GNNs in greater detail.
2. The section on expressiveness has been updated with a clearer notion of distinguishability. Theorem 3 in particular has been rephrased to be more concise.
3. We added more experiments on datasets from the long-range graph benchmark [1] to the main experiments. An additional experiment on counting subgraphs in synthetic graphs has also been added to the appendix.
4. The discussion on asymptotic runtime and table of physical runtime (now Table 8, previously Table 5) have been expanded with more details.

Additionally, we also fixed typos, broken references and other minor issues that were kindly pointed out.

We will be happy to answer any further questions and incorporate additional feedback.


[1] Dwivedi et al., Long range graph benchmark, Advances in Neural Information Processing Systems, 2022

---

### Decision · Action_Editors · 2023-06-30

**Recommendation:** Accept with minor revision

**Comment:**

Five referees recommend accept or lean accept, indicating that the contribution has significant merits as it proposes a different architecture with sufficiently convincing theoretical and empirical results. I particularly commend the authors' efforts addressing the reviewers' concerns in the rebuttal and during the discussion period.

At the end of the discussion period a few items were identified which could or should still be improved. Hence I am recommending accept with a minor revision. I request that the final version of the manuscript implements any remaining items promised during the discussion period and carefully considers the final reviewers' comments, particularly the items listed below.

* Reviewer 3MCo finds that the proof of Theorem 1 needs more explanations.
* Reviewer 9Kh9 has a remaining concern about the existence of a polynomially complex f needed in one of the proofs. While the authors provided some arguments on the existence, this was not sufficiently clear and needs to be discussed more clearly in the final version.
* Reviewer UkCv finds that several points can be improved, particularly the comparison with random-walk-based GNNs https://openreview.net/forum?id=vgXnEyeWVY&noteId=WsQGRytLre.

**Audience:**

The topic and findings are of interest to a good number of individuals in TMLR's audience.

**Claims And Evidence:**

Main claims are sufficiently supported by convincing and clear evidence. A few remaining requests are indicated below.

---

> ### Author Response · Authors · 2023-07-18
> **Revised Version Submitted**
>
> We thank the Action Editors and Reviewers for the helpful feedback.
>
> A revised version of the manuscript has just been uploaded.
> We added details on the polynomial bound of the CNN filters in the proof of Theorem 3.
> We also added additional clarifications for the remaining comments.
>
>
> When would we be expected to upload a camera ready version? Should we already do so or wait for a review of the revised version?